

# Dating Late Pleistocene pluvial lake shorelines in the Great Basin, USA using rock surface luminescence dating techniques: developing new approaches for challenging lithologies

Christina M. Neudorf[1, 2], Teresa Wriston[2], Geraint T.H. Jenkins[3], Sebastien Huot[4]

[1]Vicus, Brisbane, QLD, 4106, Australia
[2]Division of Earth and Ecosystem Sciences, Desert Research Institute, Reno, 89512, United States
[3]Coventry University, Coventry, CV1 5FB, United Kingdom
[4]Illinois State Geological Survey, Champaign Illinois, 61820, United States

*Correspondence to*: Christina M. Neudorf (c.neudorf@vicus.net.au)

**Abstract.** This study examines the feasibility of dating pluvial lake beach ridges using rock surface luminescence dating techniques. Dating pluvial lake highstands in the internally drained Great Basin of the US helps us understand the timing of changes in precipitation and temperature patterns in western North America during the Late Pleistocene. The majority of highstand ages have relied on few radiocarbon ages of shell and/or charcoal sometimes coupled with luminescence dating of sand. Within our study area in the south-central Great Basin, luminescence ages of sand-size particles have successfully dated aeolian influxes of sand during arid intervals, but have not successfully dated the highstand beach ridges, the best preserved of which are largely gravel.

Directly dating when these gravel clasts were last exposed to sunlight via luminescence is ideal but their limestone and volcanic lithologies prove challenging. Initial measurements from these lithologies show promise. Polymineral extracts from limestone clast surfaces from Coal Valley that contain sufficient detrital sediment exhibited infrared (IR) signals with low to moderate fading rates and properties suited to single-aliquot regenerative (SAR) dose measurement protocols. Ages calculated using the minimum dose model straddle the C-14 age estimate of the Pluvial Lake Coal highstand with one age consistent with the C-14 at 1σ.

Crushed slices from volcanic clasts from Cave Valley could be dated using a high-temperature (290°C) post-infrared infrared (PIRIR) signal with a correction for fading. Many ages obtained from volcanic clast surfaces were observed to be several thousand years younger than the independent age control of ~16-18 ka. This suggests that the volcanic rocks have been exposed to light long after the pluvial lake highstand, likely because of bioturbation, and that their most recent burial occurred in response to climatically-driven soil formation processes. Surprisingly, there is congruency between luminescence-depth profile plateau ages calculated from *inside* the volcanic rocks and independent age control. This suggests that some volcanic rocks were small enough to have been bleached throughout their entire thickness in the Late Pleistocene pluvial lake beach environment and that PIRIR signals that record the time of beach ridge formation may be preserved within the rock sub-surface.



This study develops novel dating approaches for challenging rock lithologies. Rock surface dating techniques for pluvial lake beach ridges in the Great Basin should be further developed with consideration of local bedrock type(s), clast size, sample collection and preparation methods, gravel bleaching processes in pluvial lake environments and the impact of soil development and bioturbation on study sites.

# 1 Introduction

## 1.1 Dating pluvial lake shorelines

Great Basin pluvial lake shorelines are invaluable indicators of past hydroclimate conditions and provide baseline data for atmospheric circulation models (Reheis et al., 2014). As such, accurate determinations of the depositional ages of these sediments and related landforms are critical for interpretations of paleoenvironmental conditions. Pluvial lake history is also important for reconstructing the early migration and settlement patterns of people in the western US (Adams et al., 2008). The abundant plants and animals within and near freshwater lakes and marshes have always attracted people. As these systems respond to changing climatic and topographic conditions, their geographic distribution changed in ways that influenced the early mobility of societies (Wriston, 2009). Therefore, reconstructing the timing of pluvial lake highstands, regressions, and expansion and contraction of fringing wetlands, can put archaeological sites into a paleoenvironmental context that helps explain their distribution.

Past studies applying a range of relative and absolute dating techniques to shoreline features have shown serious discrepancies between methods that are difficult to explain (Owen et al., 2007; Redwine et al., 2020). In particular, optically stimulated luminescence (OSL) dating techniques commonly yield inconsistent results with high error (Adams and Rhodes, 2019) or underestimate the expected ages of pluvial highstand beach ridge deposits by thousands of years despite pristine sampling conditions (Owen et al., 2007; Adams, unpublished data). Previous research in Coal Valley, Nevada yielded post-infrared infrared 225°C (pIRIR225) ages from a series of beach ridges that severely underestimate the expected age of the ~16,000 year 1522 m above sea level (asl) Lake Coal highstand. These were $4.34 \pm 0.5$ ka (1515.6 m asl), $3.69 \pm 0.42$ ka (1513.8 m asl) and $1.87 \pm 0.18$ ka (1521.5 m asl) (Wriston and Adams, 2020). We suspect that the medium sands (180-300 µm) sampled from the beach gravels were actually younger wind-blown sands that translocated into the initially open-work beach ridge gravels after beach ridge formation leading to an age under-estimate. This research investigates the feasibility of dating gravel-size clasts in pluvial beach ridge shorelines to remedy this. Gravels require greater energy to move and should reflect when water flow was strong enough to form the beach ridges. Our project area in Lincoln County, Nevada has beach ridges often comprised of limestone and volcanic rocks. Here, we test and outline novel sample preparation protocols for these materials and investigate the optical properties from polymineral grains extracted from them.



## 1.2 Luminescence dating rock surfaces

Luminescence dating determines the last time sediments have been exposed to sunlight or heat prior to deposition and burial and provides chronologies for archaeological and geological events (Huntley et al., 1985; Lian and Roberts, 2006; Roberts and Lian, 2015; Woor, 2022). Luminescence dating techniques are commonly applied to silt or sand grains, but in recent years, significant advances have been made in dating rock surfaces (Habermann et al. 2000; Greilich et al., 2005; Vafiadou et al., 2007; Liritzis, 2011; Simms et al., 2011; Sohbati et al., 2012; Freiesleben et al., 2015; Simkins et al., 2016; Khasawneh et al., 2019; Gliganic et al., 2021; Freiesleben et al., 2023, Ageby et al., 2024) as well as mineral grains encased in carbonate deposits, calcarenite and limestone (Rich et al., 2003; Prescott and Habermehl, 2008; Liritzis et al., 2008; 2010; Ageby et al., 2023).

Luminescence ages from rocks are important for sites that lack adequate sand/silt for traditional luminescence dating techniques, in addition to sites that are contaminated by mobile fine grain materials that severely post-date (underestimate) the true age of the landform or archaeological site. Rock surface dating approaches applied to natural deposits, such as beach ridges (Simms et al., 2011), flood gravels (Smith et al., 2023), or moraines (Rades et al., 2018; Yang et al., 2024), most commonly target cobble size clasts, which are sampled under dark or dim red light conditions. Sampling may be done at night and/or with a light-safe tarp or tent to block any ambient sun, moon or traffic/city light that may reset the OSL signal. Once in the lab, ~10 mm diameter cores are extracted from the rock, which are then sliced into sub-millimeter slices, and the OSL signal is measured from each slice. This produces a luminescence-depth profile that traces the OSL signal intensity from the rock surface to depth, where the depth of light penetration may be inferred from past bleaching episodes (Fig. 1). Figure 1 shows theoretical luminescence-depth profiles that would be measured after: i) light exposure of the rock surface, ii) re-burial a rock surface after an exposure event, and iii) after no light exposure or burial of the rock for extensive time. For a rock that had sufficient sun exposure prior to burial, the time of the most recent exposure event can be calculated from the near-surface plateau of the luminescence-depth profile (red line in Fig. 1).





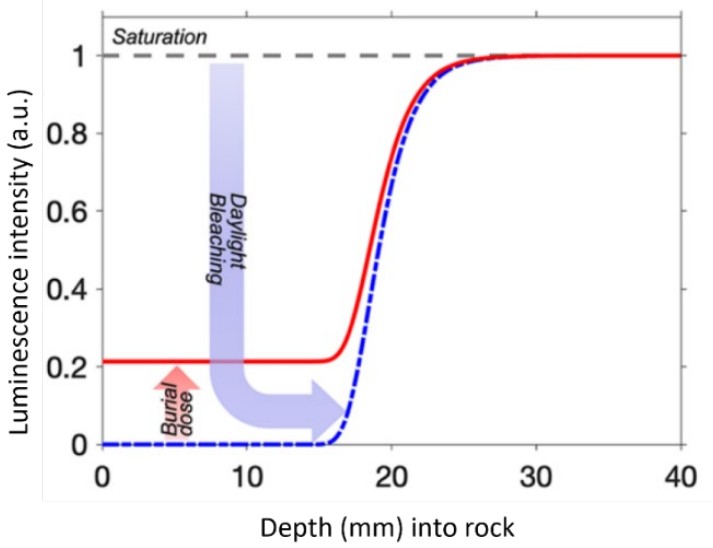

**Figure 1. Luminescence-depth profiles that are expected after i) light exposure of the rock surface (s-shaped blue dashed line), ii) re-burial a rock surface after an exposure event (solid s-shaped red line), and iii) after no light exposure (straight grey dashed line). From Smith et al. (2023).**

Generating an age from rock luminescence-depth profiles requires an estimate of the environmental dose rate at the rock

surface, as well as at depth. Dose rate models, therefore, take into account measured dose rates from the rock and surrounding sediments, and use established beta and gamma attenuation factors to calculate dose rates at depth within the rock (e.g., Jenkins et al., 2018; Riedesel et al., 2020).

## 2 Study area and sample sites

This study examines the optical and dosimetric properties from clasts collected from pluvial lake beach ridge gravels in

Coal, Cave, and Lake valleys, in Lincoln County, NV (Fig. 2, Figs S1-S4). The study area is characterized by an arid to semi-arid climate with ~25 cm of annual precipitation, a quarter of which falls during the summer. Great Basin conifer woodland occupies elevations above 1770 m, while desert scrub, including rabbit brush (Chrysothamnus nauseosus, Chrysothamnus viscidiflorus) green ephedra (Ephedra viridis), horsebrush (Tetradymia canescens) and sagebrush (Artemisia nova) dominate elevations of 1500 to 2000 m (Spaulding, 1985). Only sheer cliffs and playa floors are devoid of plants.

Coal Valley, Cave Valley, and Lake Valley all contained pluvial lakes (Pluvial Lakes Coal, Cave, and Carpenter, respectively; Mifflin and Wheat, 1979) and marsh systems during the late Pleistocene around 16,000 years ago (cf. Wriston and Adams, 2020). Warming and drying of these lakes between 14,600 and 12,900 years ago (Rhode and Adams, 2016) was stalled with cooler conditions during the Younger Dryas (ca. 12,900 to 11,700 cal yrs BP). Beginning ca. 8,000 years ago (Wriston, 2009), extreme drought dried the lakes, marshes, and springs and wind displaced much of the lake and marsh

sediments in the basin floors throughout the Great Basin. The declines in area and productivity of basin wetlands varied



spatially and temporally, likely with decadal or century-level fluctuations. The pace and tempo of the lake and wetland decline to modern conditions has yet to be fully reconstructed.

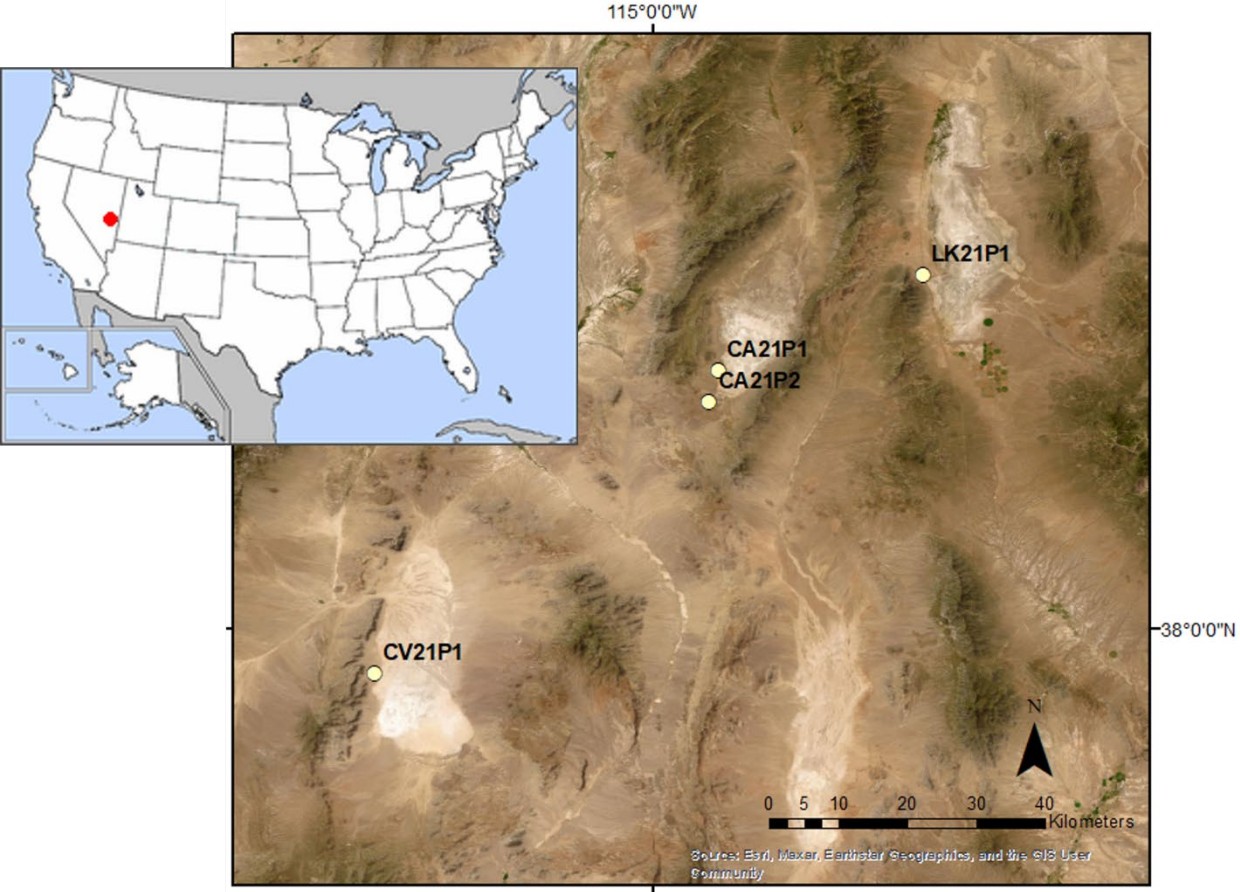

**Figure 2. Study site locations in Coal, Cave and Lake valleys, Lincoln County, NV. Basemap: ESRI Maxer, Earthstar Geographics, and the GIS User Community.**

We sampled beach ridges inferred to have been created during the highstand in each basin (Table 1). Geochronological and geomorphic evidence from Coal, Cave, and Lake valleys suggests that pluvial lakes in these basins reached their last highstand sometime between ~15,000 and ~18,000 years ago. Wriston and Adams (2020) radiocarbon dated the Coal Valley highstand to 15,873-16,281 cal yrs BP using a young freshwater Lymnaeidae Stagnicola sp. shell from backbar lagoonal deposits. The species and size of the dated shell suggest that it is unlikely to suffer from reservoir effects due to the acquisition of old carbon (cf. Pigati et al., 2010). A radiocarbon age of shoreline features from Lake Valley suggests that Lake Carpenter was near its highstand at 16,938-17,649 cal yrs BP (Currey in Lillquist, 1994; Munroe and Laabs, 2013). Duke and Young (2018) report ages of wetland sediments in the lake floor by 12,630 cal yrs BP, indicating that the lake had receded before this time and further drying occurred before a 9,270 cal yrs BP soil formed. The highstand at Cave Valley has never been directly dated; however, GIS computer simulations by Duke and King (2014) that predict the relative chronology



of lake desiccation using basin geomorphology and hydrology, suggest that Cave Lake desiccated before neighboring Lake Carpenter in Lake Valley. During recession of Cave Lake, the distributary system entering Cave Valley from the north would have expanded providing patchy wetland environments likely utilized by people during the Paleoamerican period (Duke and Young, 2018). Dated sediments from this distributary system suggest that it dried up shortly after ca. 9,400 years ago after a mesic period and low lake transgression ca. 11,300 ± 1,000 years ago based on luminescence dating of quartz grains below a gravel bar (Duke and Young, 2018).

**Table 1. Study site locations in Coal, Cave and Lake Valleys.**

| Study site | Basin | Lat/Long (decimal degrees) | Beach ridge elevation[1] (m asl) | Lake highstand elevation[1] (m asl) | Highstand age[2] (cal years BP) |
|---|---|---|---|---|---|
| CV21P1 | Coal Valley | 37.940906, -115.363785 | 1520.50 | 1522 | 15,873-16,281 |
| CA21P1 | Cave Valley | 38.337317, -114.914635 | 1851.66 | 1852 | ~16,900-18,000 |
| CA21P2 | Cave Valley | 38.296278, -114.925958 | 1851.68 | 1852 | ~16,900-18,000 |
| LK21P1 | Lake Valley | 38.461389, -114.646172 | 1826.82 | 1826 | 16,938-17,649 |

[1] Highstand elevations in meters above sea level are interpreted from highest shoreline features in satellite imagery using 1/3 arc second Digital Elevation Model from the USGS National Map (nationalmap.gov) with accuracy of around 0.82 m. The CV21P1 location was surveyed to within 20 cm corrected elevation using a Trimble Nomad during previous work (Wriston and Adams, 2020: 55).

[2] The radiocarbon age dates a time when Lake Carpenter (Lake Valley) was near (4 m below) its highstand.

Bedrock gravel material types in the study area are dominated by limestones, volcanics, and tuff. These lithologies are representative of the bedrock geology of much of Nevada (Fig. 3) (Crafford, 2007). Limestones, sandstones, and shales formed when Nevada was a broad carbonate shelf during the Paleozoic era. Younger Mesozoic and Cenozoic volcanic flows, tuffaceous rocks, intrusive igneous and sedimentary rocks formed during a subsequent period of complex tectonic movements, volcanic activity, and terrane accretion (Fig. 3) (Dickinson, 2006).





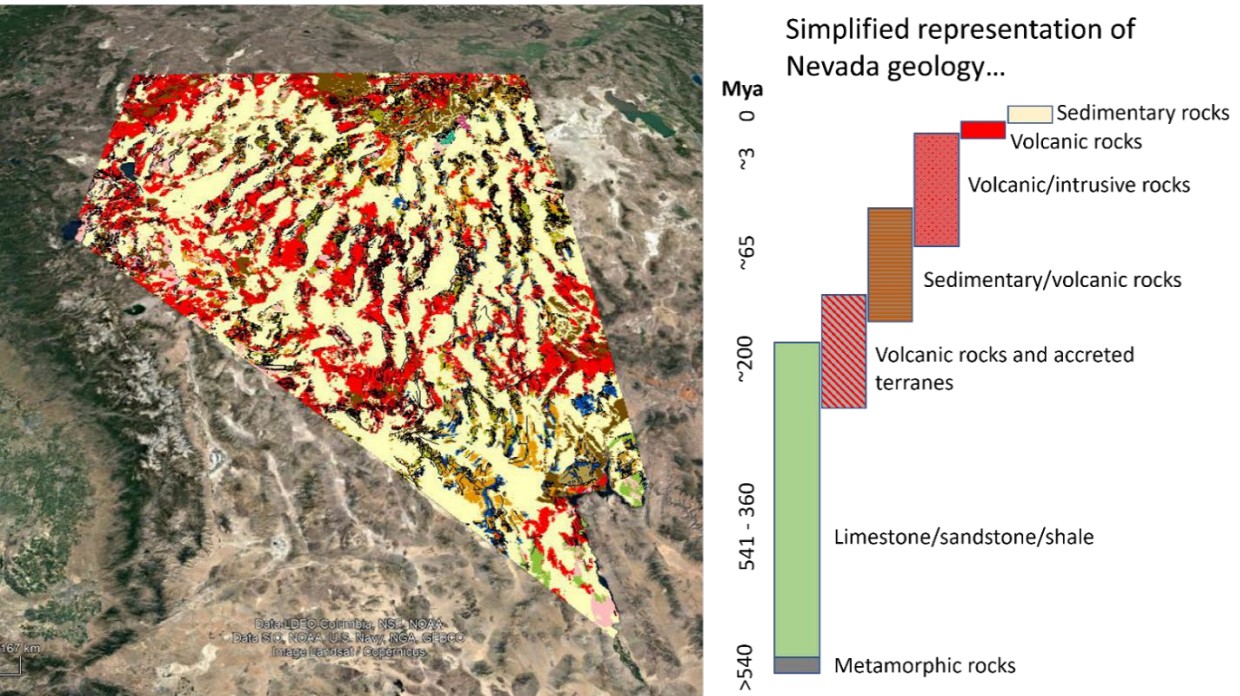

**Figure 3. Nevada surficial geology map from Crafford (2007), scale 1:250,000 (left), and simplified representation of Nevada geology (right).**

### 2.1 Coal Valley

The sampled beach ridge at 1520.5 m above sea level in Coal Valley (site CV21P1) is superimposed on a Quaternary alluvial
fan on the western side of Coal Lake basin and is the highest evident shoreline (Fig. S1). This beach ridge was created after
the last glacial maximum (MIS 2) and previously dated to ca. 16,000 (15,873-16,281 cal yr BP at 2-sigma calculated using
Calib 8.2; 13,366 ± 59 RYBP; D-AMS 029792) by Wriston and Adams (2020). It has been incised by a drainage channel
exposing sandy beach gravels overlying lagoonal silts and clays. The soils in this area have been mapped as aridisols and
entisols of the Ursine-Cliffdown Association that are typified by thin A horizons of very gravelly sandy loam that grade into
B or C horizons of gravelly sandy loam (National Cooperative Soil Survey, USA, 2025). The Golden Gate Range west of the
CV21P1 sample location is comprised of Lower Mississippian limestones, such as the Joana, Mercury, and Bristol Pass
formations underlain by Pilot shale that tops the upper Devonian Devils Gate limestones belonging to the Guilmette
Formation (Rowley et al. 2017; Tschaz and Pampayan, 1961; 1970) (Fig. S5).

### 2.2 Cave Valley

In Cave Valley, two beach ridges were sampled, one on the west side of the basin at 1851.66 m elevation (CA21P1; Fig. S2)
and one at the southwestern end of the basin at ~1851.68 m elevation (CA21P2; Fig. S3). The pluvial lake highstand is at
1852 m (Table 1). The ridge on the west side of the basin is superimposed on a Quaternary alluvial fan and is the highest of a



series of at least six ridges stepping up from the playa. The sampled beach ridge to the south (CA21P2) is wider and flanks a wavecut bench eroded into andesitic and basaltic bedrock. Both localities are dominated by aridisols with gravelly sandy

loam A horizons overlying gravelly sandy loam B horizons (National Cooperative Soil Survey, USA). Tschaz and Pampayan (1961; 1970) classify the geology above both sample areas as undifferentiated older volcanics with pockets of Pennsylvanian limestone (Fig. S6). Further west of CA21P1, in the Egan Range, Scotty's Wash quartzite and Chainman shale of upper Mississippian age outcrop and these materials may be found in the pluvial lake reworked alluvial fans near the sample areas (Fig. S6).

**2.3 Lake Valley**

In Lake Valley, the highest observed beach ridge, superimposed on a Quaternary alluvial fan and bisected by HWY 93, was sampled on the west side of the basin at ~1826.82 m asl (Table 1; Fig. S4). The pluvial lake highstand averages around 1826 m asl. Soils in the area are dominated by aridisols with gravelly ashy loam and sandy loam A horizons overlying B horizons of a similar texture (National Cooperative Soil Survey, USA, 2025).

According to mapping by Tschaz and Pampayan (1961 and 1970), the closest bedrock source to the LK21P1 sample site is middle to upper Devonian Guilmette formation Dolomite and Devils Gate Limestones exposed on the north side of Dutch John Mountain (northern margin of the Fairview Range) west of the sample area (Fig. S7). Upslope of this formation is an unconformity topped by upper Devonian Pilot Shale that has another unconformity topped by lower Mississippian Bristol Pass limestones and others.   To the northwest of the LK21P1, volcanic rocks (older, undifferentiated) and upper

Mississippian Scotty Wash quartzite, sandstone, and shale outcrops amongst the Quaternary age alluvium emanating from the Schell Creek Mountain Range (Fig. S7).

**3 Methods**

**3.1 Sampling in the field**

Sampling at all three sites was conducted at night with the aid of dim red (>660 nm) headlamps. Sample location coordinates

were recorded using a survey grade Trimble Geo7X Handheld GNSS System with external antenna. The SW facing natural exposure at Coal Valley was sampled near midnight using a pitched tarp to block light from a partial moon that set at 10:45 pm. The moon shade was also used in the secluded Cave Valley. The sample location in Lake Valley was near a highway and so an open bottomed tent covered with two layers of 6 mil black plastic was used to prevent light exposure.

At Coal Valley, the surface of the natural exposure in the drainage was cleaned, photographed and the sedimentary

characteristics recorded. Then at night, the exposure was dug back at least 5 cm to expose rocks that were unlikely to have seen daylight since burial, and 30 of the largest observed gravel clasts were collected at measured depths of approximately ~1 m below the top surface of the ridge.



At Cave Valley, 20 of the largest observed gravel clasts were collected between 38 and 44 cm from the ground surface in an excavated pit at site CA21P1. A bulk sample (~300 g) of the gravel and matrix was also collected for water content and dosimetry measurements. At site CA21P2, ~100 of the largest observed gravel clasts were collected between 33 and 64 cm depth below the surface, along with another bulk gravel matrix sample for water content and dosimetry.

At Lake Valley, ~100 of the largest observed gravel clasts were collected from ~0.5 m depth below the surface, along with a ~300 g bulk dosimetry sample. At Cave and Lake Valley sites, the orientation of most rocks were recorded by labeling their top sides with masking tape and marker after wrapping. This was undertaken to allow measurements from the clast upper faces, as these have been shown to record sub-surface bleaching more effectively than the lower face of the clasts (Jenkins et al., 2018). The orientation of some clasts could not be recorded if they fell out of the exposure prior to collection. The sediments in all pits were photographed and their sedimentary characteristics recorded in daylight after sampling was complete.

Given the diversity of rock lithologies present within bedrock surrounding Coal, Cave and Lake valleys, we expected to collect a wider variety of rock types from the sampled beach ridges (e.g., sandstones, siltstones, limestones, intrusive and extrusive igneous rocks). We found, however, that gravels at each beach ridge were dominated by the lithology of the most proximal bedrock outcrop (Figs S5-S7). Thus, our sample processing and dating protocols presented below were developed for only for the limestone and volcanic rocks that we collected.

## 3.2 Sample preparation for Equivalent dose (De) measurement

Rock sample preparation for limestone and volcanic gravels required different approaches. The volcanic gravel clasts were cored and sliced following more traditional rock surface dating techniques (e.g., Jenkins et al., 2018), however, given that luminescence signals must be measured from quartz or feldspar minerals and not from carbonates, we prepared the limestone rocks in a manner similar to that of Liritzis et al. (2010) (see below). Minerals extracted from both rock types were not further separated to isolate quartz and feldspar (e.g., Aitken, 1998) due to the small quantities of material available.

### 3.2.1 Limestones

Sample preparation for limestone rocks from Coal and Lake Valley included the following steps:

1. The outer secondary carbonate coatings were filed away with a file or Stylo-style Dremel tool.
2. The outer 1-2 mm of each side of the limestone clast was dissolved in weak (10%) hydrochloric (HCl) acid over several hours while monitoring the change in limestone thickness with repeated caliper measurements.
3. Detrital sediment released from the outer 1-2 mm of dissolved limestone (step 2) was collected, wet sieved and dried for luminescence measurements.
4. The majority of the remaining limestone was dissolved in high-concentration (~36%) HCl, and the detrital clastic sediment was collected, wet sieved and dried for signal testing and dose recovery tests.



5. The remaining residual limestone rock (~10-20 g) was dried, milled into a fine powder and submitted to ALS
Minerals, Reno, NV for U, Th, K and Rb determination using ICP-MS (U, Th, Rb) and ICP-AES (K) (see
       Section 3.6).

This approach to sample preparation for limestone gravel-sized rocks requires the following assumptions:

1. During beach ridge formation, light penetrated the outer 2 mm or more of the limestone surface to bleach the
   signals from detrital quartz and feldspar minerals.

2. The entire surface of the limestone rocks in the swash zone of the beach were adequately light exposed prior to
     burial to bleach near-surface quartz/feldspar detrital grains.

3. Limestone surfaces experienced limited or no erosion or dissolution after beach ridge formation and prior to
   sampling.

4. Micro-beta dosimetry effects that can lead to scatter in measured grain equivalent dose (De) and age
distributions are minor in the limestone and around it.

5. Any U-series disequilibrium that may have existed within the limestone during its formation will have
   corrected itself (i.e., U-238 daughters with half lives up to ~1600 years will have reached a new equilibrium
   with the parent) since formation of the limestone during the Devonian.

### 3.2.2 Volcanics

Polymineral grains (i.e., samples that had undergone no mineral separation) were extracted from volcanic gravel-sized rocks
in two ways: by coring and slicing (the traditional method), in addition to removal of the entire outer ~1 mm layer from all
sides of each rock. This second approach maximized the amount of material that could be used for De measurement:

1. The outer carbonate coating on the volcanic rocks was removed by setting them in a bath of high-concentration
   (36%) HCl acid.

2. The rocks were cored using a 10 mm inner-diameter diamond core drill bit. Cores penetrated the entire rock
     where possible, and where possible, multiple (2-3) cores were extracted from rocks of adequate size. Where the
     orientation of the rocks were known, rocks were cored on their top surfaces through to their bottom surfaces.

3. Rock cores were sliced into ~1 mm thick slices using a ~0.3 mm thick wafer blade mounted on a low-speed
   precision cutting saw. The polymineral slices were subsequently crushed gently by hand using an agate mortar
and pestle and sieved into distinct grain size fractions between 125 and 250 µm for measurement.

4. The outer ~1 mm layer of the remaining rock was removed using diamond bur bits mounted on a variable-
   speed Dremel tool.

5. The 32-63 µm polymineral fraction of the dremeled outer ~1 mm rock layers were extracted using suspension
   settling and Stokes Law, then dried for measurement.

Our sample preparation protocol for volcanic gravels requires the following assumptions:





1.  Minerals extracted by coring and slicing require that at least one side (the top or bottom) of the rock was exposed to sunlight long enough prior to burial to yield accurate ages.

2.  Minerals extracted from the outer ~1 mm layer of all sides of the rock require that all sides of the rock were adequately exposed to sunlight prior to burial.

## 3.3 Luminescence measurement

In this study polymineral samples from limestone clasts were measured at the single-grain level, while samples from volcanic clasts were measured using multi-grain aliquots. Multi-grain aliquots of polymineral fractions were mounted onto 10 mm diameter stainless steel discs using silicon oil as an adhesive. Aliquot diameters were 3 mm and their luminescence signals and De measurements were measured in one of two Risø TL/OSL-DA-20 readers equipped with a 90Sr/90Y beta radiation source. For single-grain measurements, single grains were mounted onto single-grain discs, each containing one hundred 300 μm diameter holes. Single grains were measured in the same Risø readers using IR lasers.

Tests were conducted to determine whether or not quartz minerals from the samples yielded a datable luminescence signal (Section 4.2). For these measurements blue light stimulation was made with a cluster of blue LEDs (NICHIA type NSPB-500AS) with a peak emission at 470 nm and a total power of ~80 mW/cm2 and signals were detected with a bialkali EMI 9235Q photomultiplier tube (PMT) fitted with Hoya U-340 filters that transmit UV light. For feldspar signal measurements from multi-grain aliquots of polymineral sediment, IR stimulation was made with a cluster of Vishay TSFF 5200 IR diodes with peak emission at 870 nm, and maximum power of 115 mW/cm2 at the sample position. Single-grain feldspar signal measurements were made with an IR laser emitting 830 nm at a maximum power of 150 mW/cm2. Corning 7-59 and Schott BG39 filters were used for K-feldspar signal detection. An additional Schott RG780 filter was mounted in front of the IR laser to reduce background noise during single-grain measurements (Lai et al., 2002).

## 3.4 Equivalent dose (De) determination

### 3.4.1 SAR

The equivalent dose (De) was measured using variations of the single aliquot regenerative dose (SAR) protocol (Wallinga et al., 2000; Murray and Wintle, 2000). This protocol measures the sensitivity-corrected natural signal (Ln/Tn) of each grain/aliquot, followed by sensitivity-corrected regenerative dose signals (Lx/Tx) measured after a series of increasing laboratory doses (i.e., the dose response curve). The protocol included the measurement of a repeat-dose point (i.e., one regenerative dose point is measured twice) to measure the recycling ratio, and a zero-dose point (i.e. Lx/Tx is measured after no dose is given) to measure recuperation. Aliquots/grains were rejected from further analysis if the recycling ratio was beyond 10% of unity and if recuperation was greater than 5% of the sensitivity-corrected natural signal. Aliquots/grains were also rejected if their natural test dose signals were less than three times the standard deviation of the background signal.





We applied varying SAR protocols during this study to measure the infrared signal measured at 50°C (IR50 signal) as well as post-infrared infrared (pIRIR) signals measured at a range of temperatures. These protocols are shown in Tables 2 and S1.

**Table 2. The IR50 and pIRIR290 protocols used in this study.**

| Step | $IR_{50}$ (multi-grain) | $IR_{50}$ (single-grain) | $pIRIR_{290}$ (multi-grain) |
|---|---|---|---|
| 1 | Natural/Regenerative Dose | Natural/Regenerative Dose | Natural/Regenerative Dose |
| 2 | Preheat (160°C, 10 s) | Preheat (160°C, 10 s) | Preheat (320°C, 10 s) |
| 3 | IR diodes (50°C, 200 s) → $Ln, Lx$ | IR laser (50°C, 2 s) → $Ln, Lx$ | IR diodes (50°C, 100 s) |
| 4 | Test dose (~6 Gy) | Test dose (~6 Gy) | IR diodes (290°C, 100 s) → $Ln, Lx$ |
| 5 | Preheat (160°C, 10 s) | Preheat (160°C, 10 s) | Test dose (~10 Gy) |
| 6 | IR diodes (50°C, 200 s) → $Tn, Tx$ | IR laser (50°C, 2 s) → $Tn, Tx$ | Preheat (320°C, 10 s) |
| 7 | IR diodes (180°C, 100 s) | IR diodes (180°C, 100 s) | IR diodes (50°C, 100 s) |
| 8 | Return to step 1. | Return to step 1. | IR diodes (290°C, 100 s) → $Tn, Tx$ |
| 9 | | | IR diodes (325°C, 40 s) |
| 10 | | | Return to step 1. |


### 3.4.2 Dose recovery tests

In this study, SAR protocols were tested prior to De measurement using dose recovery tests (Aitken, 1994; Roberts et al., 1999). During these tests the sample was bleached using IR or pIRIR stimulation, administered a known laboratory dose to the sample, then measured using the SAR protocol to be tested. A protocol was deemed suitable for the sample if the ratio of 290 the measured-to-given dose was within 10% of unity and few aliquots or grains were rejected from analysis due to dim signals, high recuperation or recycling ratios that are not within 10% of unity.

### 3.5 Anomalous fading

Fading rates were measured using a SAR procedure modified from Auclair et al. (2003). This procedure entails bleaching the sample, administering a known laboratory dose, preheating the sample to a temperature deemed appropriate by the dose 295 recovery tests, then measuring the sensitivity-corrected signal (Lx/Tx) after a series of delay times ranging from a few minutes to ~10 h.

In this study we measured fading rates from several multi-grain aliquots and/or single grains from each sample, where the fading rate from each aliquot/grain is illustrated using a fading plot of Lx/Tx versus delay time in hours (h). Delay time is plotted on a logarithmic scale to account for the fact that feldspar signals fade exponentially. The fading rate of an 300 aliquot/grain is quantified by the slope of the line, termed the g-value, with units of percent per decade, where a decade is a 10-fold increase in delay time (i.e., one increment on a logarithmic scale) (Huntley and Lamothe, 2001).





## 3.6 Dosimetry

The dose rates of the sampled clasts and the surrounding bulk matrix samples were determined in two ways: i) using measured concentrations of parent radionuclides Uranium (U), Thorium (Th) as well as Potassium (K) and Rubidium (Rb)

using inductively coupled plasma mass spectrometry (ICP-MS) (for U and Th) and inductively coupled plasma atomic emission spectroscopy (ICP-AES) (for K), and ii) using high resolution gamma spectrometry to measure K-40, as well as U-238 and Th-232 radionuclides and their daughter products. This second approach allows us to assess the possibility of secular disequilibrium within the U or Th decay chains that may have led to fluctuations in environmental dose rates through time (Murray et al., 1987; Ivanovich and Harmon, 1992).

Both gravel matrix and gravel rock subsamples from all sites were prepared for ICP-MS and ICP-AES analysis of radionuclide contents by ALS Geochemistry, Reno, NV. These samples were dried and milled to a fine, flour consistency and subsamples used for U, Rb and Th measurement were fused with lithium borate and measured with ICP-MS. $K_2O$ was measured from the bulk sample with ICP-AES and converted to % K.

Gravel matrix samples from sites CV21P1 (Coal Valley) and CA21P1 (Cave Valley) were measured using high resolution

gamma spectrometry, as well as one sample containing a mix of subsamples from multiple representative limestone gravel rocks from CV21P1. Gamma spectrometry measurements could not be made for individual limestone gravel rocks, because they did not form large enough samples needed (~50 g) for measurement. Samples were dried, ground to a fine powder consistency, then ashed at 450 °C in a muffle furnace for 24 hours. They were then emplaced in γBeakers designed by miDose Solutions, Poland to prevent leakage of Rn gas and stored for a minimum of 21 days to allow the sample to reach

equilibrium. After storage, samples were placed on top of a broad energy, planar High Purity Germanium (HPGe) detector and the gamma emission was measured for 7 days. The gamma energy spectrum was analyzed to calculate the activities of U-238, Ra-226, Th-232, Pb-210, Ra-224, Ac-228 and K-40 in Bq/kg.

Dose rates (Gy/ka) were calculated in the Dose Rate and Age Calculator (DRAC) by Durcan et al. (2015) using the conversion factors of Liritzis et al. (2013) and assumed negligible water contents within the gravels and an average water

content of 3 ± 5 % within gravel matrices. The cosmic dose rate (Gy/ka) was calculated according to Prescott and Hutton (1994) using sample geographic coordinates and depths below the surface.

## 4 Results

### 4.1 Beach ridge sedimentology

#### 4.1.1 Coal Valley

Sediments exposed at site CV21P1 reveal two main units including a massive, cohesive blocky silt (Unit 1 – clayey silt) overlain by a weakly bedded gravel with medium-coarse sandy matrix (Unit 2 – sandy beach gravel) (Fig. 4(a), Table 3). Gravel clasts rarely exceed 40 mm along their intermediate (b-) dimension and lithologies are dominated by limestones



derived from outcropping Devonian carbonate shelf deposits to the west (Fig. S5). These sediments are interpreted to represent pluvial lake highstand beach gravels overlying fine-grained sediments that were ponded in a lagoon behind the beach ridge shortly before rising waters deposited the gravels over them (Wriston and Adams, 2020) (Fig. 5). In this interpretation, the gravels and underlying lagoonal deposits are contemporaneous (cf. Adams and Wesnousky, 1998). Therefore, the age of radiocarbon dated shell from the lagoonal deposits (15,873-16,281 cal yrs BP) is interpreted to represent the age of the beach gravels that were deposited there during the pluvial Lake Coal highstand.





(a)

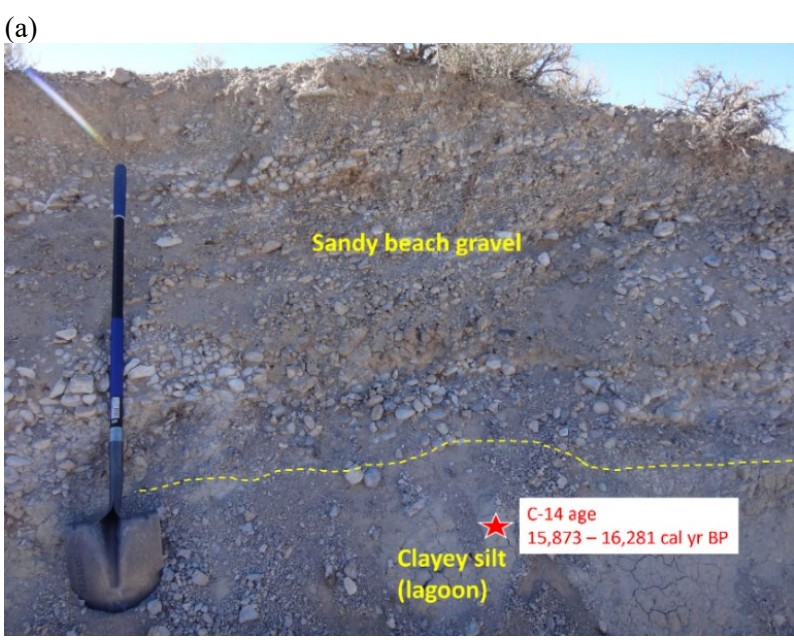

(b)

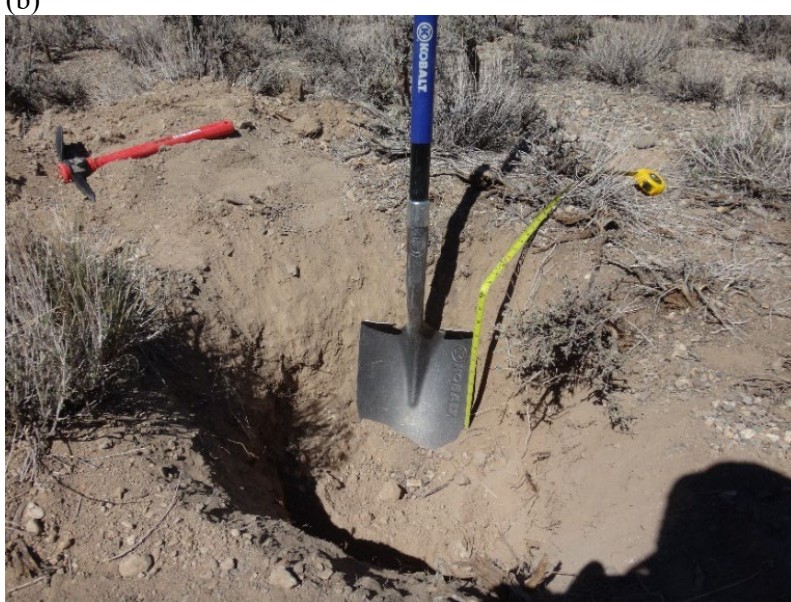

**Figure 4. (a) Sampled sedimentary exposure CV21P1 in Coal Valley. (b) Sampled excavation pit from site CA21P1.**



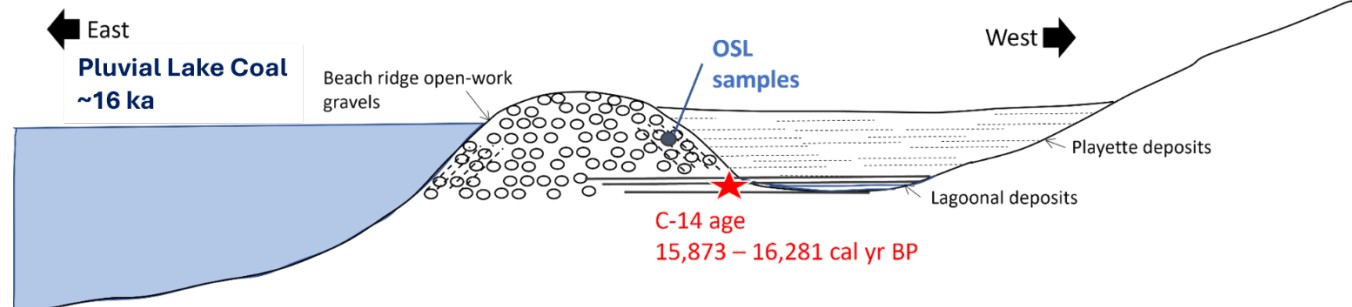


**Figure 5. Illustration of sampled beach ridge shortly after formation ~16,000 years ago during the Pluvial Lake Coal highstand. Any backwater playette deposits or lagoonal deposits landward of the beach ridge have since eroded away, exposing the beach gravels on the west side of the ridge. Figure not to scale.**

**Table 3. Sedimentary characteristics of beach ridge at site CV21P1.**

| Depth below surface (cm) | Sedimentary unit | Description | Interpretation | Samples collected |
|---|---|---|---|---|
| 0-120 | Unit 2 | Weakly bedded gravel with medium-coarse sandy matrix capped by very thin A horizon. Root penetration most prevalent in upper 7 cm. Gravel clasts dominated by rounded-subrounded weakly imbricated limestone with apparent dip N. Gravel shapes are tabular, equant and oblong with b-axes up to ~4 cm. Rare orange andesite gravels present. Moderately sharp lower contact that dips to the N. | Sedimentation in a near-shore, high-energy swash zone. | 30 gravels from ~1 m below the surface for luminescence testing. |
| 120- base of exposure | Unit 1 | Massive, cohesive carbonaceous blocky silt. | Back-beach lagoonal sediments deposited under low-energy conditions. | Freshwater shell with C-14 age of 15,873-16,281 cal yrs BP |



### 4.1.2 Cave Valley

Sediments exposed in excavation pits at sites CA21P1 and CA21P2 include gravel with silty sand matrix. Sedimentary structures were not visible due to the loose silt that quickly covers exposed sections. Both beach ridge gravels are dominated by volcanic rock lithologies derived from outcropping Miocene and Oligocene flows and tuffaceous sedimentary rock to the
west and south, with more felsic varieties appearing more common at CA21P1 (Figs 4(b), S6). Gravel clasts were subangular to subrounded with their longest dimensions commonly under 10 cm.

### 4.1.3 Lake Valley

Sediments exposed in the excavation pit of site LK21P1 include gravel with silty sand matrix. Limestone dominates the rock lithologies, which is derived from neighboring Devonian lithified carbonate shelf deposits (Fig. S7). Gravel clast sizes were
similar to those found in Cave Valley. Samples collected from Lake Valley were not datable (see below), so further discussion of the site sedimentology is not reported here.

### 4.2 Luminescence signals from limestone and volcanic rocks

Luminescence signals were measured from polymineral samples extracted from a subset of limestones from Coal Valley and Lake Valley as well as volcanic rocks from Cave Valley to determine their brightness and suitability for dating.
Measurements were made in both continuous-wave (CW-OSL) and linear-modulation (LM-OSL) modes (details in Supplementary Material, Section 3). Because none of the minerals extracted from five Lake Valley limestone gravels had detectable luminescence signals, we focused our efforts on samples from Coal and Cave valleys. Neither limestone samples nor volcanic rocks contained quartz with a datable fast component, but some rocks from both lithologies exhibited IR signals. Volcanic rocks with the brightest IR signals were typically of the felsic (andesitic or rhyolitic) varieties common at
site CA21P1, whereas basalts, like those collected from site CA21P2, had no signal.

### 4.3 Limestone gravels from Coal Valley

### 4.3.1 Dose recovery tests

Twenty-one limestone rocks from site CV21P1 were processed for IR signal testing and De measurement (Table 4). Nineteen of these were clasts excavated within the beach ridge at ~1 m depth, and two were cobbles collected from the
ground surface (i.e., modern samples). Minerals extracted from six of these gravels passed dose recovery tests either at the multi-grain or single-grain level, but only three of these (rocks #2, 10, and 18) had sufficient material left for De measurement and enough aliquots/grains that passed SAR rejection criteria. Minerals extracted from the two cobbles collected from the ground surface (rocks #2M and 3M, Table 4) did not have a detectable IR or OSL signal.



**Table 4. Dose recovery (DRT) and fading test results for polymineral sand grains extracted from limestone rocks from Coal Valley site CV21P1. MG refers to multi-grain analysis and SG refers to single-grain analysis. The IR50 SAR protocol (Table 2) was applied.**

| Sample | Thickness[1] (mm) [mass (g)] | Passed MG DRT? | Passed SG DRT? | MG fading rate (%/decade) | SG fading rate (%/decade) | Notes |
|---|---|---|---|---|---|---|
| Ancient samples | | | | | | |
| Rock 1 | 18.5 (0.043) | - | Yes | - | - | Dim, not enough grains pass rejection criteria |
| Rock 2 | 27.8 (0.229) | Yes | Yes | 3.2 ± 0.2 (n=4) | 2.6 ± 0.5 (n=65) | Datable |
| Rock 3 | 26.1 (0.771) | - | - | 3.3 ± 0.6 (n=3) | | No signal. |
| Rock 5 | 32.5 (0.048) | Yes | No | 0 (n=3) | - | No accepted grains at SG level. Insufficient material |
| Rock 7 | 41.9 (0.081) | - | - | - | - | No signal |
| Rock 8 | 27.8 (0.007) | - | - | - | - | Not enough sediment to measure |
| Rock 9 | 21.5 (0.171) | - | Yes | 9.9 ± 1.0 (n=4) | - | Very low # of accepted grains |
| Rock 10 | 32.0 (0.800) | - | Yes | - | 2.6 ± 1.7 (n=28) | Datable |
| Rock 11 | 21.2 (0.890) | Yes | Yes | - | - | Low number of accepted grains |
| Rock 13 | 34.0 (0.163) | - | - | - | - | No signal |
| Rock 14 | 20.8 (0.007) | - | - | - | - | No signal |
| Rock 16 | 18.5 (0.030) | - | - | - | - | Dim signal, no grains pass rejection criteria |
| Rock 17 | 20.4 (0.633) | - | - | - | - | No signal |
| Rock 18 | 21.9 (0.604) | - | Yes | - | 2.0 ± 0.7 (n=27) | Datable |
| Rock 19 | 20.8 (0.031) | - | No | - | - | Not enough grains pass rejection criteria |
| Rock 20 | 21.9 (0.392) | No | - | - | - | No signal |
| Rock 21 | 35.9 (0.012) | - | - | - | - | Too few grains with a signal |
| Rock 22 | 18.5 (0.054) | - | - | - | - | Too few grains with a signal |
| Rock 23 | 34 (1.132) | - | - | - | - | No signal |
| Modern samples | | | | | | |
| Rock 2M | 46.0 (0.831) | - | - | - | - | No signal |
| Rock 3M | 83.3 (0.985) | - | - | - | - | No signal |

[1]Thickness refers to the shortest rock dimension and mass is the mass of polymineral grains extracted from the outer 1-2 mm of the rock.




Dose recovery tests for limestone rocks #2, 10 and 18 were conducted using the IR50 signal and the single-grain approach (Table 2). Because the available grain size fraction from the rocks was significantly smaller (63-250 μm) than the width of the single-grain disc holes (300 μm) in the reader, it is likely that several (up to 5) grains within each disc hole contributed to the IRSL signal, and the results should be viewed as small multi-grain aliquot data, or "micro-hole" data as termed by Berger

et al. (2013). Natural grain signals were depleted using an IR LED stimulation of each disc for 1000 s followed by IR laser stimulation (2 s) of each grain, then each disc was administered a ~21 Gy beta dose before being measured using SAR. The results are plotted in Figure S9. Measured sigma-b values were 22%, 18% and 17% where sigma-b refers to measurement uncertainties that are attributed to instrument reproducibility and grain-to-grain variations in signal properties (Galbraith and Roberts, 2012). Measured-to-given dose ratios were $0.99 \pm 0.02$, $0.91 \pm 0.03$ and $0.96 \pm 0.03$ for Rocks #2, 10 and 18,

respectively suggesting that the IR50 SAR protocol is suitable for the Coal Valley limestone samples.

### 4.3.2 Fading rates

Fading measurements for limestones were made on the same grains (or small multi-grain aliquots) used for De determination. These included the 63-90 μm fraction from Rock 2, the 125-180 μm fraction from Rock 10 and the 180-250 μm fraction from Rock 18. This allows each grain/small multi-grain aliquot to be corrected for its own fading rate, which can

vary significantly from grain to grain or aliquot to aliquot. An average fading rate was also measured from four 2 mm diameter multi-grain aliquots from the 90-125 μm grain size fraction from Rock #2.

Representative single-grain/small aliquot IR50 signals and fading plots are shown in Figure S12. The fading rate, or g-value, of each grain/aliquot was measured using multiple prompt and delayed signal measurements. The weighted mean g-value for Rock 2 (63-90 μm fraction) is $2.55 \pm 0.52$ %/decade, which is slightly lower but within 2 standard deviations of that

measured from the 90-125 μm grain size fraction from the same rock ($3.19 \pm 0.15$ %/decade). The weighted mean g-value for Rocks 10 and 18 is $2.65 \pm 1.71$ and $1.99 \pm 0.79$ %/decade, respectively.

### 4.3.3 Radionuclide concentrations

ICP-MS/AES measurements of U, Th, K and Rb were conducted on crushed and milled subsamples of Rocks 2, 10 and 18 as well as a bulk sample of the gravel matrix (Table S6). All radionuclide contents for the limestone gravels are low relative to

values typically obtained for bulk non-carbonaceous sediment, and this is expected given the relatively low concentration of silicate minerals within the limestone. The gravel matrix yielded values that are slightly above those of the rocks, reflecting a higher silicate mineral content.

HPGe measurements of the limestone yielded results that are in range of those measured using ICP-MS/AES, while measurements made from the gravel matrix deviate somewhat from the ICP-MS/AES results. To check the calibration of our

HPGe detector, measurements were made on a standard prepared by Murray et al. (2015) that was measured by 23 different laboratories (Table S7, Fig. S17 A&B). Our results are within 20% of those previously published.



Radionuclide activity ratios for the U and Th series in the limestone and gravel matrix samples are reported in Table S10 and plotted in Figures S17 (C-F) and 18. The gravel matrix sample from CV21P1 shows elevated Pb-210 activity relative to U-238 and Ra-226, while the Ra-226/U-238 ratio approximates unity and the Ra-224/Ac-228 ratio falls within 20% of unity.

Given that the gravel matrix sample was collected from loose, porous medium-coarse sandy gravels ~1 m below the surface, we interpret the elevated Pb-210 values to reflect the translocation of atmospheric Pb-210 that has leached down to the level of sampling during rain events (Murray, 1996). The Ra-226/U-238 and Ra-224/Ac-228 ratios from the gravel matrix otherwise suggest that the deposit approximates secular equilibrium. Radionuclide activity ratios calculated from the limestone rocks are well within 20% of unity, suggesting that our assumption that any U-disequilibrium that may have

existed within the limestone during its formation will have corrected itself since Devonian times (Section 3.2.1).

### 4.3.4 Dose rate modelling

Alpha, beta and gamma dose rates have been calculated for all limestone rocks as well as the gravel matrix from site CV21P1 (Table 5). Dose rates were calculated from both the ICP-MS/AES and HPGe measured radionuclide concentrations. As expected for carbonate-rich materials, all dose rates are low. HPGe measurements from limestone rocks yielded alpha,

beta and total dose rates within the range of results obtained from ICP-MS/AES analysis. HPGe measurements of the gravel matrix underestimate the ICP-MS/AES beta and gamma results leading to a ~21% reduction in the total dose rate.

Dose rates were calculated for the outermost ~2 mm of each limestone taking into account the measured alpha, beta and gamma dose rates from each rock, beta and gamma dose rates from the surrounding bulk gravel matrix as well as cosmic rays from outer space. Dose rates for each rock were modeled using the approach of Riedesel and Autzen (2020), which

incorporates experimentally derived attenuation factors for granite, assuming that the rock is shaped like a sphere. This approach yields total dose rates that are lower than those calculated using earlier approaches that assume laterally infinite beta and gamma dose rates for both the rock and surrounding sediment (e.g., Jenkins et al., 2018) (see Section 9 of Supplementary Material for a comparison). To check that the beta and gamma attenuation factors of Riedesel and Autzen (2020) are appropriate for our limestone samples, the elemental concentrations of 4 representative limestone subsamples

were measured by XRF and used to calculate attenuation factors using Geant4 simulations. These simulations generated attenuation factors that matched those published by Riedesel and Autzen (2020) for granite (Autzen pers. comm.), and so we apply them to our dose rate models in our study.

Modeled alpha, beta and gamma dose rates with depth into Rocks 2, 10 and 18 are shown in Figure 6. These are based on radionuclide concentrations measured by ICP-MS/AES (Section 4.3.3). Similar dose rate with depth models were also

calculated using the radionuclide contents determined using HPGe. The total dose rate used for calculating the age of detrital mineral grains within the outer 2 mm layer of each limestone rock is an average value highlighted in Figure 6.





**Table 5. Calculated gravel and sediment dose rates for site CV21P1.**

| Rock sample | Grain size [1] (µm) | Method [2] | Alpha [3] (Gy/ka) | Beta [3] (Gy/ka) | Gamma [3] (Gy/ka) | Cosmic (Gy/ka) | Total [4] (Gy/ka) |
|---|---|---|---|---|---|---|---|
| Rock 2 | 90-125 | ICP/AES | 0.05 ± 0.01 | 0.28 ± 0.01 | 0.02 ± 0.01 | 0.25 ± 0.03 | 0.94 ± 0.04 |
| Rock 2 | 63-90 | ICP/AES | 0.07 ± 0.02 | 0.28 ± 0.03 | 0.02 ± 0.01 | 0.25 ± 0.03 | 0.96 ± 0.05 |
| Rock 10 | 125-180 | ICP/AES | 0.05 ± 0.02 | 0.14 ± 0.01 | 0.01 ± 0.01 | 0.24 ± 0.02 | 0.78 ± 0.04 |
| Rock 18 | 180-250 | ICP/AES | 0.02 ± 0.01 | 0.20 ± 0.02 | 0.01 ± 0.01 | 0.25 ± 0.03 | 0.85 ± 0.04 |
| Gravel matrix | 125-180 | ICP/AES | 0.11 ± 0.03 | 0.64 ± 0.07 | 0.42 ± 0.04 | 0.25 ± 0.03 | 2.06 ± 0.13 |
| Gravel matrix | 125-180 | HPGe | 0.10 ± 0.04 | 0.35 ± 0.02 | 0.28 ± 0.02 | 0.25 ± 0.03 | 1.62 ± 0.12 |
| Crushed limestone | 125-180 | HPGe | 0.06 ± 0.02 | 0.25 ± 0.01 | 0.18 ± 0.02 | 0.25 ± 0.03 | 1.38 ± 0.11 |

[1] Dose rates for gravel matrices are reported here for the 125-180 µm grain size fraction only.

[2] Method of radionuclide measurement. ICP-MS (ICP) was used to measure U and Th contents, and ICP-AES (AES) was used to measure K contents by ALS Minerals, Reno, NV. HPGe measurements were used to obtain U, Th and K contents at DRILL.

[3] External alpha, beta and gamma dose rates have been corrected for grain size and water content. Alpha dose rates assume an a-value of 0.15 ± 0.05 following Balescu and Lamothe (1994).

[4] Total dose rates include an internal beta dose rate component that assumes an internal K content of 12.5 ± 0.5 % for felspar grains following Huntley and Baril (1997). Dose rate models for gravels below calculate dose rates for the grain size fraction measured for dating and accounts for dose rate attenuation rates with depth into the rock.





(a) Rock 2

(b) Rock 10

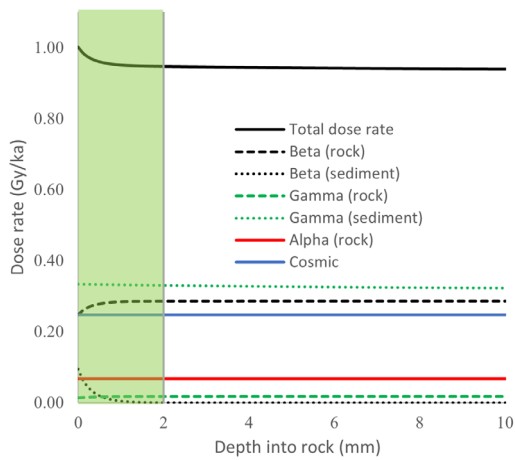

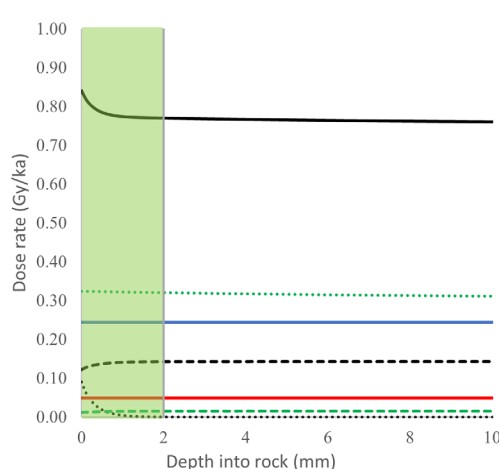

(c) Rock 18

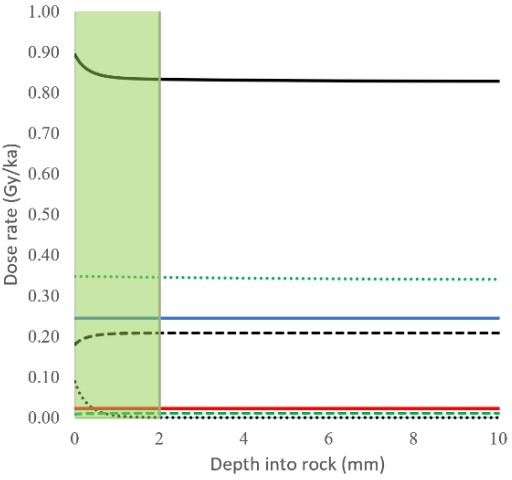

**Figure 6. Modeled dose rates for limestone rocks from site CV21P1 following the method of Riedesel and Autzen (2020) and radionuclide concentration data obtained from ICP-MS/AES. The average dose rate for the outer ~2 mm of the rocks (green shading) was used for age calculation.**

### 4.3.5 Limestone luminescence age distributions

The De and luminescence ages were measured from polymineral grains from Rocks 2, 10 and 18; these rocks passed dose recovery tests (Section 4.3.1). The De was measured from two grain sizes from Rock 2 (63-90 μm and 90-125 μm fractions), the 125-180 μm fraction from Rock 10, and the 180-250 μm fraction from Rock 18. De was measured using the IR50 SAR protocol in Table 2 for single-grains, and each grain (or small multi-grain aliquot) was corrected for its own fading rate using the correction model of Huntley and Lamothe (2001) for all samples except for the 90-125 μm fraction from Rock 2. All aliquots from the 90-125 μm fraction of Rock 2 were corrected for fading using an average fading rate measured from



medium sized multi-grain aliquots (Section 4.3.2). As Figure S13 shows, single-grain fading corrections have the effect of magnifying single-grain age errors for older grains within the distribution, and this is attributed to the limited precision with which we can measure single-grain fading rates.

The fading-corrected aliquot age distributions are shown in Figure 7 where the cumulative distribution plot is superimposed on a kernel density estimate (KDE) curve. Overdispersion values (OD) for the fading-corrected data are reported in Table 6. These have been calculated using the central dose model of Galbraith et al. (1999) after taking into account the measured sigma-b value in dose recovery tests (Section 4.3.1) and record the level of spread in the data that can be attributed to environmental factors during burial, such as incomplete re-setting of grain signals by sunlight or heterogeneities in the dose

rate field that lead to grain-to-grain variations in acquired dose (Galbraith and Roberts, 2012).

Given that most rock dating studies derive ages from rock (primarily granite, sandstones, quartzites, and volcanics) slices, rather than single-grains or small aliquots, the relationship between single-grain/aliquot age distribution shapes, OD, and depositional process for rocks is not known and has not been examined in the same way these relationships have been examined for traditional sediment dating studies (e.g., King et al., 2014). We provide preliminary interpretations of these

data that should be solidified by future testing. The OD of the 90-125 μm fraction of Rock 2 is 30%, which may indicate that most grains have been sufficiently bleached prior to burial (cf. Arnold et al., 2009, table 4 for quartz). The OD of all other samples are much higher, ranging from 65% to 98%, and may record incomplete bleaching as well as scatter that is the result of micro-beta dosimetry effects. KDE curves for the fading-corrected data are nearly symmetrical for Rock 10, possibly indicating that most scatter is due to micro-beta dosimetry effects (Maya et al., 2006), while those of Rocks 2 and 18 show a

slight positive skew that may record micro-beta dosimetry effects as well as incomplete bleaching of grains.

### 4.3.6 Age models

Statistical models have been developed for calculating a sample De value from a distribution of single-grain De values (e.g., Galbraith et al., 1999; Galbraith and Roberts, 2012; Guérin et al., 2017). These models are based on assumptions regarding the depositional history and composition of the sample analyzed. Single-grain ages in this study were calculated using the

minimum dose (MDM) and central dose (CDM) models of Galbraith et al. (1999) as well as the more recently developed average dose (ADM) model of Guérin et al. (2017). A description of the assumptions behind each model are outlined in the Supplementary Material, Section 10.

### 4.3.7 Age results

All calculated ages are summarized in Table 6 and plotted in Figures 7 and 8 alongside the radiocarbon age for the

freshwater shell collected from the lagoonal silts (Unit 1). The C-14 age plots closest to the MDM ages for all limestone samples, while the CDM and ADM ages most severely over-estimate the expected age. The C-14 age agrees with the MDM ages from the 90-125 μm fraction of Rock 2 at 1 σ, and at 2σ, the C-14 age agrees with the 63-90 μm fraction of Rock 2 as well as the CDM and MDM ages of Rock 18 calculated using the HPGe-derived dose rates (Table 6).



Figure 8 shows the relationship between ages determined using ICP-MS/AES dose rates and those determined using HPGe

dose rates. These results must be viewed with caution as the HPGe measurements could not be made from each individual

limestone sample, but rather were made from one milled sample that combined limestone pieces from multiple rocks from

the site. The method of dose rate determination impacted the oldest ages of those compared, where the ICP-MS/AES method

resulted in younger age estimates relative to the HPGe method.





Figure 7. CAM, MAM and ADM model results superimposed on the age distribution KDE plot. The C-14 age for site CV21P1 is indicated as a dashed red line. Ages have been calculated using dose rates determined through ICP-MS/AES. *Single-grain fading-corrections have been applied to all datasets accept for the 90-125 µm fraction from Rock 2, where all aliquots have been corrected using an average g-value obtained from multi-grain aliquot fading measurements (Section 4.3.2).



**Table 6. Table 6. Calculated ages (at 1σ) for limestone gravels from site CV21P1 in Coal Valley. The calibrated 2σ radiocarbon age obtained from lagoonal deposits underlying the gravel gravels is 15,873 – 16,281 cal yr BP. Luminescence age values that agree with the C-14 age within 1σ are highlighted in bold; those ages agreeing with the C-14 age at 2σ are bold and italicized.**

| Limestone # | Grain size (μm) | Type of *De* measurement[1] | Type of g-value measurement[1] | Dose rate (Gy/ka) | OD (%) | Modeled age (ka) | | |
|---|---|---|---|---|---|---|---|---|
| | | | | | | ADM age | CDM age | MDM age |
| Dose rates determined through ICP-MS & ICP-AES | | | | | | | | |
| Rock 2 | 90-125 | SG | MG | 0.94 ± 0.04 | 30 ± 6 | 23.5 ± 1.5 | 22.5 ± 1.5 | **15.3 ± 1.9** |
| Rock 2 | 63-90 | SG | SG | 0.96 ± 0.05 | 65 ± 12 | 73.9 ± 12.7 | 59.9 ± 7.3 | ***29.6 ± 6.8*** |
| Rock 10 | 125-180 | SG | SG | 0.78 ± 0.04 | 81 ± 17 | 51.2 ± 8.2 | 36.8 ± 7.6 | 9.0 ± 2.7 |
| Rock 18 | 180-250 | SG | SG | 0.85 ± 0.04 | 96 ± 20 | 52.0 ± 15.1 | 32.8 ± 7.8 | 7.4 ± 2.4 |
| Dose rates determined through HPGe gamma spectrometry | | | | | | | | |
| Rock 2 | 90-125 | SG | MG | 0.82 ± 0.04 | 30 ± 6 | 26.8 ± 1.8 | 25.6 ± 1.7 | **17.3 ± 2.2** |
| Rock 2 | 63-90 | SG | SG | 0.85 ± 0.05 | 71 ± 12 | 91.4 ± 15.9 | 71.3 ± 9.1 | ***28.3 ± 6.2*** |
| Rock 10 | 125-180 | SG | SG | 0.79 ± 0.04 | 81 ± 18 | 50.9 ± 8.6 | 36.5 ± 7.8 | 8.6 ± 2.5 |
| Rock 18 | 180-250 | SG | SG | 0.78 ± 0.03 | 98 ± 20 | 61.4 ± 20.5 | ***30.8 ± 9.0*** | ***11.3 ± 3.5*** |

[1] SG = single-grain, MG = multi-grain aliquot. Because the available grain size fraction from the rocks was significantly smaller (63-250 μm) than the width of the single-grain disc holes (300 μm), it is likely that more than one grain within each hole contributed to the IRSL signal, and the results should be viewed as small multi-grain aliquot data, or "micro-hole" data as termed by Berger et al. (2013).







**Figure 8. CAM, MAM and ADM ages plotted with the C-14 age from Unit 1 that are all from the same Coal Valley profile CV21P1. Ages determined using ICP-MS/AES dose rates are plotted against those determined using HPGe dose rates. The radiocarbon age limits are delineated in red and represent 1σ. Luminescence age error bars are plotted at 1σ.**



### 4.4 Volcanic gravels from Cave Valley

#### 4.4.1 Dose recovery tests

Preliminary dose recovery tests were conducted on 3 mm diameter multi-grain aliquots from all volcanic rocks from Cave Valley that exhibited an IR signal. These were all collected from site CA21P1. The rocks were ground and sieved into 32-63 μm grain size fractions prior to measurement. Each aliquot was bleached using a pIRIR regenerative dose (Lx) measurement (Table 2), then administered a ~40 Gy beta dose prior to measurement using SAR. The pIRIR protocols tested included the 180 °C, 225 °C and 290 °C protocols (Tables 2, S1), and results obtained from both the IR50 and pIRIR signals from each protocol were plotted (three aliquots per rock) (Fig. S10). Despite some outlying values of the measured-to-given dose, most aliquots passed the dose recovery test suggesting that feldspars in these rocks were suitable for SAR.

After assessment of volcanic rock fading rates (Section 4.4.2) and rock characteristics (i.e., the composition of Rock 14, below), additional dose recovery tests were conducted on Rocks 4, 7, 11, 12, 13 and 18 using a minimum of 24 multi-grain aliquots per rock. These tests were conducted on the ground and suspension-settled 32-63 μm grain size fraction and were used to calculate a sigma-b value (i.e., measurement uncertainties that are attributed to instrument reproducibility and grain-to-grain variations in signal properties (Galbraith and Roberts, 2012)). IR50 and pIRIR290 signals from all rocks passed dose recovery tests, yielding measured-to-give dose ratios within 10% of unity and sigma-b values equal to 12% or less (Fig. S11). Cutting and coring of Rock 14 revealed that most of this rock was composed of cryptocrystalline quartz (chert). This chert does not have a luminescence signal, so Rock 14 was not processed further.

#### 4.4.2 Fading rates

Measurements of anomalous fading were conducted on the 32-63 μm fractions tested above. As expected, fading rates were the highest for the IR50 signal (up to ~40 %/decade), and generally decreased with increasing pIRIR signal temperature (Fig. S15). The pIRIR290 signal yielded an average g-value of 3.3 ± 0.7 %/decade, while the pIRIR180 and IR50 signals yielded average g-values of 10.3 ± 0.9 and 23.1 ± 1.8 %/decade, respectively. This implies that the highest temperature pIRIR signal (pIRIR290) may be used for dating after a fading correction, and lower temperature signals (e.g., pIRIR180 and IR50) have such high fading rates that they cannot be used for dating.

#### 4.4.3 Radionuclide concentrations

ICP-MS/AES and HPGe measurements of volcanic Rocks 4, 7, 11, 12, 13 and 18 are shown in Table S9. As expected, radionuclide concentrations are much higher than those observed in limestone (Section 4.3.3). The HPGe results in Table S9 agree with the ICP-MS/AES results within 1 sigma for U and K, and within 2 sigma for Th. Radionuclide activity ratios for U in the gravel matrix collected from site CA21P1 show elevated levels of Pb-210 relative to U-238 that is likely atmospheric and sourced from recent rain events (Table S10, Fig. S19) (Murray, 1996). All other activity ratios fall within 20% of unity, suggesting that disequilibrium in the U and Th decay chains is limited or negligible.





### 4.4.4 Dose rate modelling

Calculated dose rates for the gravel matrix from site CA21P1 are high, with a total dose rate equal to ~5 Gy/ka (Table 7). Dose rates calculated for all volcanic rocks using ICP-MS/AES radionuclide concentrations are very high, ranging from ~7 to ~9 Gy/ka (Table 7). Total dose rates include an internal beta dose rate component that assumes an internal K content of 10 ± 2 % for feldspars following Smedley et al. (2012), but we acknowledge that this value may be high as K contents of more intermediate or andesitic rocks are typically lower than 10% (Taylor, 1968). Electron dispersion microscopy (SEM-EDS) was used to measure semi-quantitative relative elemental concentrations of volcanic rock multi-grain aliquots, and K was detected in relatively small concentrations in all rocks (Figs. S20-S22). Dose rates calculated for rock surfaces and rock age-depth profiles below are corrected for attenuation according to the sieved grain size fraction measured for dating (Brennan et al., 1991). Alpha, beta, gamma and total dose rate attenuation within each rock was modeled using the approach of Riedesel and Autzen (2020). Dose rate attenuation with depth is plotted in Figure 9.

Scanning electron microscope (SEM) photographs of grain surfaces (Figs S20-S22) suggest that while many grains may be coherent mineral crystals, it is possible that many sieved grains may actually be clumps of smaller grains that would lead to uncertainties in our dose rate attenuation corrections. Feathers et al. (2019) made similar observations from fine-grain volcanically sourced rocks from Peruvian geoglyphs. He concluded that feldspar grains did not exceed 40 μm and applied an internal K concentration of 6 ± 3%. Given the uncertainties regarding feldspar internal K content and the grain size of volcanic feldspar grains in our study, dose rates and fading-corrected ages were re-calculated for all rock slices from Rock 4 assuming: i) a smaller grain size range (10-40 μm), ii) an a-value of 0.030 ± 0.002 after measurements from volcanic rocks by Feathers et al. (2019), and iii) an internal K content of 2.8 ± 0.3 % based on ICP-AES measurements from Rock 4 (Table S9). The re-calculated rock slice IR50 ages increased by an average of 3.4 ± 1.6% (n=30) (or 230 years), and the pIRIR290 ages increased by an average of 3.5 ± 0.8% (or 423 years), suggesting that grain size and internal K content assumptions made here are a small (5% or less) source of uncertainty in our calculated ages.



**Table 7. Calculated gravel and sediment dose rates for site CA21P1.**

| Rock sample | Grain size [1] (µm) | Method [2] | Alpha [3] (Gy/ka) | Beta [3] (Gy/ka) | Gamma [3] (Gy/ka) | Cosmic (Gy/ka) | Total [4] (Gy/ka) |
|---|---|---|---|---|---|---|---|
| Rock 4 | 32-63 | ICP/AES | 1.00 ± 0.22 | 3.47 ± 0.01 | 2.19 ± 0.13 | 0.31 ± 0.03 | 7.14 ± 0.26 |
| Rock 7 | 32-63 | ICP/AES | 1.07 ± 0.23 | 4.39 ± 0.01 | 2.59 ± 0.16 | 0.30 ± 0.03 | 8.52 ± 0.29 |
| Rock 11 | 32-63 | ICP/AES | 1.23 ± 0.27 | 4.51 ± 0.01 | 2.77 ± 0.16 | 0.32 ± 0.03 | 9.00 ± 0.32 |
| Rock 12 | 32-63 | ICP/AES | 1.01 ± 0.22 | 4.17 ± 0.01 | 2.42 ± 0.14 | 0.31 ± 0.03 | 8.09 ± 0.27 |
| Rock 13 | 32-63 | ICP/AES | 1.20 ± 0.26 | 4.47 ± 0.01 | 2.70 ± 0.16 | 0.31 ± 0.03 | 8.86 ± 0.31 |
| Rock 18 | 32-63 | ICP/AES | 1.12 ± 0.24 | 4.42 ± 0.01 | 2.62 ± 0.16 | 0.31 ± 0.03 | 8.64 ± 0.29 |
| Gravel matrix | 32-63 | ICP/AES | 0.57 ± 0.13 | 2.47 ± 0.24 | 1.41 ± 0.12 | 0.30 ± 0.03 | 4.92 ± 0.30 |
| Gravel matrix | 32-63 | HPGe | 0.65 ± 0.14 | 2.61 ± 0.16 | 1.56 ± 0.09 | 0.30 ± 0.03 | 5.29 ± 0.24 |

[1] Dose rates for gravel matrices are reported here for the 32-63 µm grain size fraction only. Dose rate models for gravel surfaces and gravel age-depth profiles below calculate dose rates for the sieved grain size fraction measured for dating.

[2] Method of radionuclide measurement. ICP-MS (ICP) was used to measure U and Th contents, and ICP-AES (AES) was used to measure K contents by ALS Minerals, Reno, NV. HPGe measurements were used to obtain U, Th and K contents at DRILL.

[3] External alpha, beta and gamma dose rates have been corrected for grain size and water content. Alpha dose rates assume an a-value of 0.10 ± 0.05 for the 32-63 µm grain size fraction.

[4] Total dose rates include an internal beta dose rate component that assumes an internal K content of 10 ± 2 % for feldspars following Smedley et al., 2012. Dose rates reported here are for the 32-63 µm grain size fraction measured from dremeled rock surface samples. Rock dose rates calculated for crushed rock slices are adjusted to the sieved grain size fraction (125 to 250 µm) used for rock slice De measurements as well as dose rate attenuation with depth into the rock.


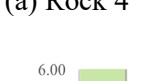



(a) Rock 4

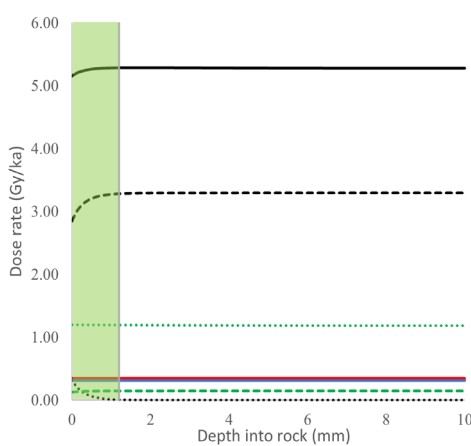

(b) Rock 7

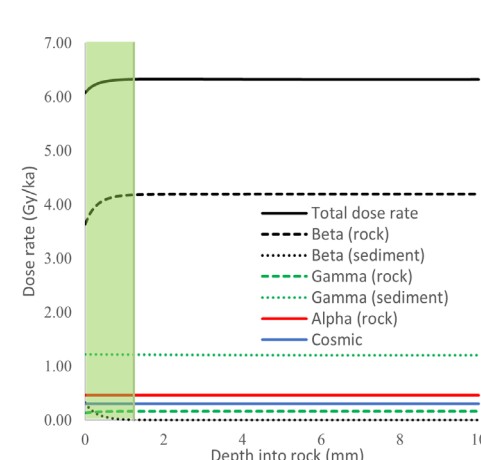

(c) Rock 11

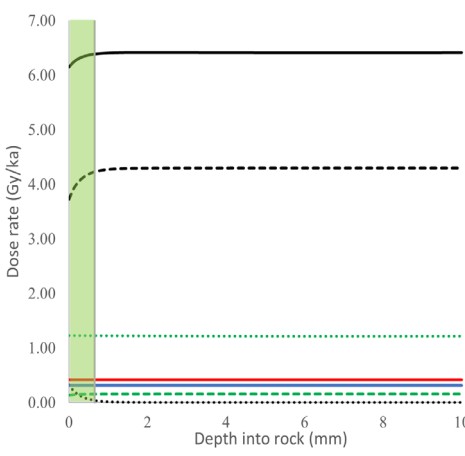

(d) Rock 12

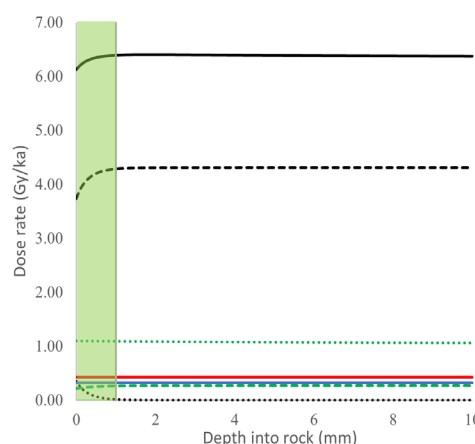

(e) Rock 13

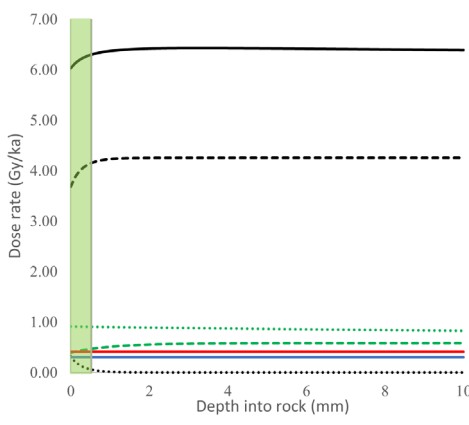

(f) Rock 18

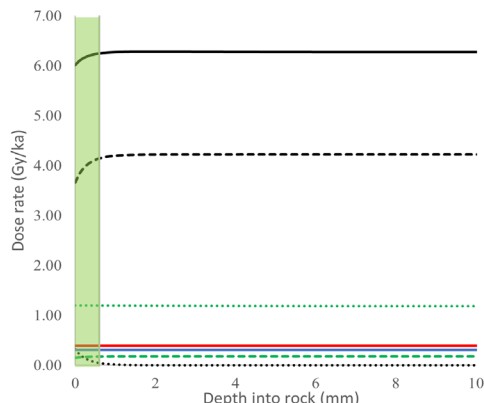





**Figure 9. Modeled dose rate with depth into gravel surfaces for rocks from CA21P1 in Cave Valley. Models are based on ICP-MS/AES measurements. The average dose rate for the outer-most slice of the rocks was calculated using the beta, gamma and alpha dose rates within the shaded green region.**





**4.4.5 Rock surface luminescence age distributions**

The outer ~1 mm layer (dremeled 32-63 μm fraction) of Rocks 4, 7, 11, 12, 13 and 18 was measured for De using the pIRIR290 protocol for multi-grain aliquots in Table 2. Measurements were made on 3 mm diameter multi-grain aliquots, each containing ~2600 grains. Aliquot De values (not corrected for anomalous fading) were divided by the average calculated dose rate within ~1 mm of the gravel surface to obtain ages for both the IR50 and pIRIR290 signals. The pIRIR290 ages were corrected for fading using the Kars et al. (2008) approach based on the model of Huntley (2006). Corrections entailed the quantification of the parameter, ρ', which is the density of recombination centers in the sample mineral (Huntley, 2006). This was estimated from multi-grain aliquot fading measurements (n=3 to 4) from each rock using equation 5 from Kars et al. (2008) and the function, analyse_FadingMeasurement() from the Luminescence R package (Kreutzer and Burow, 2023) (Table S5). Rock-specific ρ' values and aliquot-specific dose response curve (DRC) data were then used to construct unfaded and natural (simulated) dose response curves for each aliquot (dashed lines in Fig. S16) using the calc_Huntley2006() function from the Luminescence R package (King and Burow, 2023). Laboratory-measured Ln/Tn values were then interpolated onto the natural DRCs to compute fading-corrected pIRIR290 De values and ages.

As expected, IR50 uncorrected ages are significantly younger than pIRIR290 uncorrected ages (Figs 10 and 11) and this is attributed to the high rate of fading of the IR50 signal as well as the lower bleaching rate of the pIRIR290 signal. Fading-corrected aliquot pIRIR290 age distributions for the dremeled outer rock surfaces are plotted in Figs 10 and 11. Weighted mean surface ages for each rock were calculated using CDM and tabulated in Table 8.





**Figure 10. Uncorrected (left) and fading-corrected (right) age distributions from polymineral grains extracted from the outer ~1 mm layer of volcanic Rocks 4, 7 and 11 (~2500 grains per aliquot, 32-63 μm fraction) from CA21P1 in Cave Valley. Corrections apply the Kars et al. (2008) approach based on the model of Huntley (2006). Black dots are IR50 data (younger age group in each**



radial plot on the left), red dots are pIRIR290 data (older age group in each radial plot on the left, and fading-corrected ages on the right).





(a) Rock 12

(b) Rock 12

(c) Rock 13

(e) Rock 13

(f) Rock 18

(g) Rock 18

**Figure 11. Uncorrected (left) and fading-corrected (right) age distributions from polymineral grains extracted from the outer 1 mm layer of volcanic Rocks 12, 13 and 18 (~2500 grains per aliquot, 32-63 μm fraction) from CA21P1 in Cave Valley. Corrections**





**apply the Kars et al. (2008) approach based on the model of Huntley (2006). Black dots are IR50 data (younger age group in each radial plot on the left), red dots are pIRIR290 data (older age group in each radial plot on the left, and fading-corrected ages on the right).**

### 4.4.6 Rock age-depth profiles

De was measured from rock core slices using the same pIRIR290 SAR protocol that was used for volcanic rock surfaces (Figs S24, S25). Some slices could not be measured due to limited sample material or dim signals and aliquots that failed

SAR aliquot rejection criteria. Due to irregular rock surfaces and heterogeneities in rock hardness, slice thicknesses varied.

As expected, IR50 De values consistently underestimate the pIRIR290 De values in all rocks, and that includes those De values measured from the surface (i.e., top- or bottom-most) slice. Most luminescence De profiles show a decline of one or both signal De values toward at least one surface of the rock indicating limited, and heterogeneous light penetration into the rock surfaces. Where De values rise with depth, pIRIR290 De values increase at a more rapid rate than the IR50 De values,

and to a higher apparent saturated level, leading to vertically enhanced versions of the IR50 De profiles; this is typical of combined IR50 and pIRIR luminescence-depth profiles reported elsewhere (e.g., Sohbati et al., 2015; Freiesleben et al., 2015; Jenkins et al., 2018).

Aliquot slice De values were divided by depth-attenuated dose rates calculated in Section 4.4.4 to derive age-depth profiles (Figs 12-14). Age-depth profiles generally show similar patterns to those observed in the De profiles. Fading-corrected

pIRIR290 age-depth profiles further support our interpretation above that gravels experienced short-term, heterogeneous light exposure to their surfaces prior to burial. Clear, plateaus that intersect the rock surface suggestive of a long-term bleaching event prior to gravel burial (such as that shown in Figure 1) are absent.

Attempts to apply a light bleaching model to the luminescence signal depth profiles to predict pre-burial profile shapes (e.g., Freisleben et al., 2015; Khasawneh et al., 2019) were unsuccessful; due to the scatter in the data and relative paucity of data

points from each rock, model results were overly sensitive to parameter starting values. Therefore, plateaus were detected in the data using the statistical test for homogeneity of Galbraith (2003) following the approach of Gliganic et al. (2021). Using this approach, we identified statistically consistent populations of pIRIR290 ages within each slice and rejected outlying values. Then slice ages in each depth profile (each calculated using CDM) were systematically tested against each other to identify plateaus within each age-depth profile (yellow highlighted slices in Figs 12-14). Plateau ages were then calculated

by applying CDM to all statistically consistent groups of ages (Table 8).











**Figure 12. Uncorrected (left) and fading-corrected (right) age-depth profiles for volcanic Rocks 4 and 7 from CA21P1. Dashed lines denote slice thickness. Aliquots contain ~125 grains each of the 125-250 μm fraction. Top and bottom of the rock indicated for oriented samples. Yellow shading highlights slices used for plateau age calculations. Black dots are pIRIR290 aliquot ages, X symbols are rejected aliquots, and red hollow circles are CDM weighted mean ages for each slice.**

(a) Rock 11

(b) Rock 11

(c) Rock 13

(d) Rock 13

**Figure 13. Uncorrected (left) and fading-corrected (right) luminescence-depth profiles for volcanic Rocks 11 and 13 from CA21P1. Dashed lines denote slice thickness. Aliquots contain ~125-165 grains each of the 125-250 μm fraction. Yellow shading highlights slices used for plateau age calculations. Black dots are pIRIR290 aliquot ages, X symbols are rejected aliquots, and red hollow circles are CDM weighted mean ages for each slice.**



(a) Rock 18 Core 1

(b) Rock 18 Core 1

(c) Rock 18 Core 2

(d) Rock 18 Core 2

(e) Rock 18 Core 3

(f) Rock 18 Core 3





**Figure 14. Uncorrected (left) and fading-corrected (right) luminescence-depth profiles for 3 cores from Rock 18 from CA21P1.**
**Dashed lines denote slice thickness. Aliquots contain ~125 grains each of the 180-250 μm fraction. Yellow shading highlights slices**
**used for plateau age calculations. Black dots are pIRIR290 aliquot ages, X symbols are rejected aliquots, and red hollow circles are**
**CDM weighted mean ages for each slice.**

### 4.4.7 Dremmeled rock surface age results

Weighted mean pIRIR290 ages for the dremeled outer ~1 mm layer of each volcanic rock (32-63 μm grain size fraction) are summarized in Table 8 and Figure 15(a). The pIRIR290 fading-corrected ages range from ~4 to ~16 ka, where all rocks except for rocks 13 and 18 date to the mid-late Holocene between ~3 and 6 ka. When the minimum dose model is applied to all outer rock surface ages, the modeled age is statistically equivalent to the youngest rock at $3.74 \pm 0.62$ ka.

### 4.4.8 Age-depth profile results

Rock surface ages were calculated from the outer-most slices of each rock core (125-250 μm sized grains from the top, bottom or sides of the rocks), as well as from statistically consistent slice ages (the plateaus) within each age-depth profile (Table 8, Fig. 15(b)). Again, fading-corrected surface ages vary widely with pIRIR290 ages ranging from ~4.2 to ~33 ka. In most cases, the rock surface slice ages are equal to, or older than the dremeled surface ages discussed in Section 4.4.7 above (Rock 13 shows an exception to this), suggesting that rock slices may contain unbleached, residual signal as a result of irregular cut rock surfaces. Rock top and bottom age values show significant variability within and between rocks suggesting that sun exposure was not uniform across rock surfaces prior to burial and/or the rocks experienced phases of re-mobilization prior to final emplacement.

Weighted mean ages calculated from age-depth profile plateaus show more consistency from core to core in each rock as well as between rocks than the surface slice ages (Fig. 15(b)). Fading-corrected pIRIR290 plateau ages range between ~13.7 ka to ~19.2 ka for all rocks except for Rock 18. All 3 cores from Rock 18 show core-to-core consistency in plateau age values clustering between ~28 ka and 33 ka.





**Table 8. Volcanic rock fading-corrected ages from CA21P1 in Cave Valley. "Oriented" rocks were sampled with their top and bottom surfaces recorded. The expected age of the pluvial lake highstand in Cave Valley is ~16,900-18,000 years, and ages consistent with this at 1σ are in bold, while those consistent at 2σ are bold and italicized.**

| Volcanic rock # | Depth (cm) | Core length [Rock dimensions] (mm) | Surface dose rate (Gy/ka) | Dremeled surface pIRIR290 age (ka) | Surface slice(s) pIRIR290 age (ka) | Plateau pIRIR290 age (ka) |
|---|---|---|---|---|---|---|
| Rock 4 (unoriented) | 38 | 16 [55x35x16] | 5.56 ± 0.28 | 5.24 ± 0.20 | 9.64 ± 0.72<br>6.99 ± 0.81 | 13.68 ± 1.06 |
| Rock 7 (oriented) | | [70x50x14]] | | | | -- |
| Core 1 | 44 | 10 | 6.66 ± 0.39 | 3.74 ± 0.29 | **18.08 ± 0.65 (T)**<br>4.23 ± 0.41 (B) | **18.08 ± 0.65** |
| Core 2 | | 14 | | | **17.05 ± 0.5 (T, B)** | **17.05 ± 0.75** |
| Rock 11 (oriented) | 36 | 14 [70x45x14] | 6.74 ± 0.37 | 4.87 ± 0.23 | 9.00 ± 0.91 (T)<br>13.51 ± 1.50 (B) | 15.03 ± 0.64 |
| Rock 12 | 42 | [70x40x30] | 6.24 ± 0.13 | 5.35 ± 0.49 | | -- |
| Rock 13 (oriented) | 42 | 6 [70x40x6] | 6.36 ± 0.45 | *15.66 ± 0.77* | 4.88 ± 0.42 (T)<br>*19.21 ± 1.04 (B)* | *19.21 ± 1.04* |
| Rock 18 (oriented) | | [55x40x16] | | | | |
| Core 1 | 42 | 16 | 6.28 ± 0.34 | 11.13 ± 0.66 | *19.93 ± 1.82 (T)*<br>10.85 ± 1.05 (B) | 28.03 ± 1.28 |
| Core 2 | | 16 | | | **18.73 ± 2.16 (T)**<br>31.27 ± 1.07 (B) | 31.27 ± 1.07 |
| Core 3 | | 16 | | | 32.94 ± 1.42 (T)<br>*20.84 ± 1.46 (B)* | 32.94 ± 1.42 |





(a)

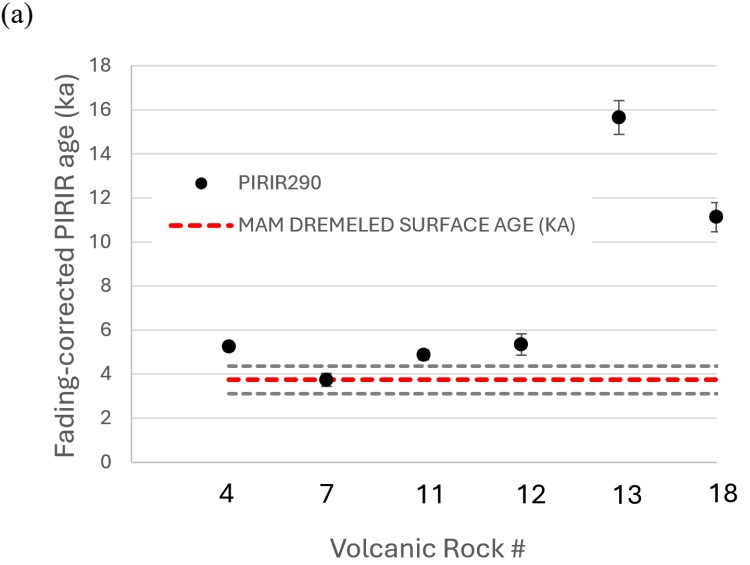

(b)

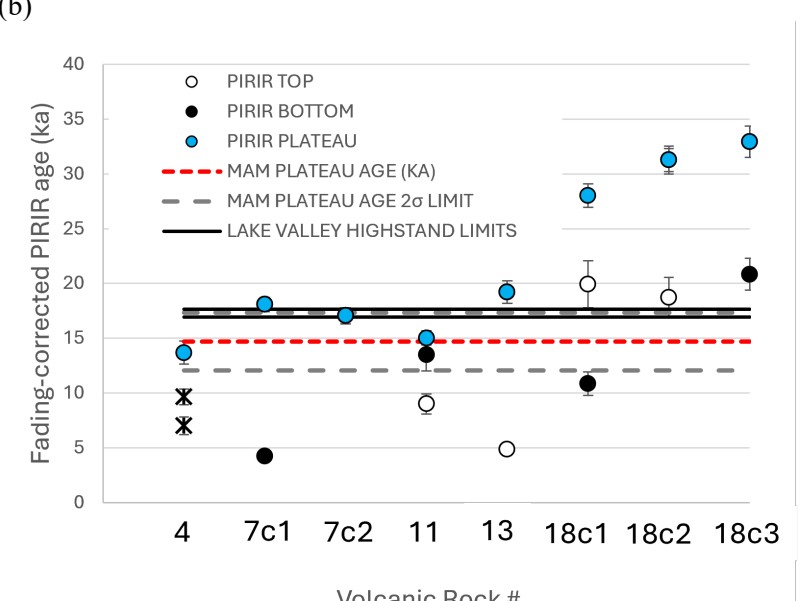

**Figure 15. (a) Weighted mean ages of CA21P1 samples calculated from 32-63 μm polymineral grains extracted from the outer ~1**
**mm of the rock surfaces using a Dremel tool. Each age is the weighted mean of 5 to 24 multi-grain aliquot ages. The red dashed**
**line marks the calculated minimum dose model (MDM) age of all rock surface ages. Dashed grey lines mark MDM age 1σ error**
**boundaries. (b) pIRIR290 rock surface slice ages and plateau ages from CA21P1 samples. Rock #4 was not oriented, so both**
**surface slices are shown as "x" symbols. The top and/or bottom of cores 11, 13, 7c2, 18c2 and 18c3 are consistent with the plateau**
**age. The 180-250 μm polymineral fraction was measured using 3 mm diameter aliquots (~125 grains per aliquot). Ages are the**
**CDM weighted mean of 1 to 3 multi-grain aliquot ages. The red dashed line marks the calculated minimum dose model (MDM)**
**age of all plateau ages. Dashed grey lines mark MDM age 2σ error boundaries. Black lines show the upper and lower C-14 age**
**limits for the Lake Valley highstand at 2σ.**



## 5 Discussion

Luminescence dating approaches applied in this study were highly dependent on the lithology of the rocks available to date.
Gravel lithologies in pluvial lake beach ridges are dominated by the lithology of the closest bedrock outcrops that provided rock for shoreline transport. These materials vary significantly from site to site and from valley to valley. In our study this necessitated the development of two very different dating approaches – one for limestone in Coal Valley and one for volcanic rock in Cave Valley. The results obtained from each site are discussed below.

### 5.1 Coal Valley

#### 5.1.1 Significance of limestone rock ages

Luminescence ages derived from limestone gravels from Coal Valley varied significantly from rock to rock and even showed dependence on measured grain size within the same rock (compare ages for grain size fractions 63-90 um and 90-125 um for Rock 2, Table 6). This variability is likely attributed to non-unform bleaching of rock surfaces, variability of rock light transmission properties (Ou et al., 2018), differences in chemical or physical weathering, as well as micro-beta
dosimetry effects that result from non-uniform rock composition (Meyer et al., 2018). Both the CDM and ADM models overestimate the age of the beach ridge, suggesting that most grains have not been completely bleached prior to burial. MDM ages straddle the C-14 age estimate of the deposit with one age consistent with the C-14 at 1σ. MDM ages that appear to post-date the time of beach ridge formation may not be derived from grains inside the limestone at all, but rather, despite our efforts to remove outer carbonate coatings, may be contaminating grains that became cemented onto the limestone
surface as part of a pedogenic carbonate coating.

#### 5.1.2 Challenges

The main challenge dating limestone clasts relate to the limited quantity of detrital grains available for dating (Table 4, column 2). Limestone formations in Nevada can vary in their detrital sediment content but are commonly associated with, or interbedded with, sandstones and shales (e.g., Rowley et al., 2017; Hurtubise and Bray, 1988). The concentration of detrital
sediment available within limestone rocks for dating varies from location to location, and this necessitates the need for testing for viable luminescence signals from each site, perhaps with some guidance from local geological maps. Quartz signals from polymineral grain extracts exhibited no fast component in any of the limestone gravels tested, so further analysis was restricted to feldspar. Grains that do have an IR signal, however, were typically well suited to SAR, and had moderate fading rates that are relatively easy to correct for, even at the single-grain level.



### 5.1.3 The single-grain advantage

For limestone gravels, it is preferable to apply the single-grain approach as opposed to measuring larger multi-grain aliquots, primarily because the single-grain approach requires relatively small amounts of material, but also because it produces a high-resolution age distribution for each sample that can be examined for evidence for incomplete bleaching of grains or other sources of scatter. Past research shows that it is common for rocks to experience only partial sun exposure prior to burial on one or more sides (e.g., Meyer et al., 2018; Smith et al., 2023), so single-grain dating, and perhaps the MDM model, should allow us to target grains that are more likely date the most recent bleaching event. However, we must keep in mind that the relationship between single-grain age distribution shapes, overdispersion (OD), rock transport and depositional history, and chemical and physical weathering of limestone rock surfaces, has yet to be examined in detail, and should be the focus of future research.

### 5.2 Cave Valley

### 5.2.1 Luminescence signals of volcanic rocks

Luminescence signals suitable for dating are most likely to be found in volcanic rocks that are intermediate to felsic in composition and contain a higher concentration of K-rich feldspars. Such types of rock are common in Lincoln County, with intermediate silicic ash flow tuffs and other tuffaceous sedimentary rocks covering more of Nevada than any other type of rock (Crafford, 2007). Intermediate and felsic volcanic rocks are best represented at site CA21P1 in our study.

### 5.2.2 Sources of variability in age-depth profiles

Volcanic rock surface ages and age-depth profiles from site CA21P1 suggest heterogeneous light exposure on rock surfaces. The variability in rock surface ages, where clast tops often appear younger or older than clast bottoms, and the variability between age-depth profiles from within the same rock, are consistent with observations made at other sites (e.g., Rades et al., 2018; Souza et al., 2019; Smith et al., 2023) and are indicative of incomplete bleaching of all rock sides prior to burial. Bleaching rates in rocks are dependent on rock surface aspect, light transmission of the rock, daylight spectrum, intensity, and duration of exposure (Ou et al., 2018; Smedley et al., 2021; Furhmann et al., 2022).

Variability in age-depth profiles may also be attributed to rock composition. Rocks that are light in color, and fine-grained, with a homogeneous mineral composition over the scale of analysis, are anticipated to be better and more uniformly bleached than rocks that are dark in color, coarse-grained, and have a heterogeneous composition over the scale of analysis (Ou et al., 2018; Meyer et al., 2018). Heterogeneities in the dose rate field within gravels and near gravel surfaces could also lead to micro-beta dosimetry effects that are not accounted for in our dose rate models.



### 5.2.3 Evidence of soil accretion after the pluvial lake highstand

The oldest ages of the dataset from Cave Valley approach ~33 ka (Rock 18) (Table 8). This supports the inference that the
beach ridge at site CA21P1 formed no earlier than MIS 2. However, an unexpected finding of this study was the large
number of anomalously young ages obtained from volcanic gravel surfaces from site CA21P1. Gravel surface ages obtained
from pIRIR290 signals suggest that many gravel surfaces were light exposed long after the last pluvial lake highstand in
Cave Valley, and even as recently as the late Holocene (Table 8, Fig. 15). Given that our sampling depth at this site was less
than 0.5 m below the present-day surface, this must be due, in part, to i) bioturbation, and possibly ii) post-burial light
exposure of gravels in what were initially open-work gravels near the beach ridge surface after beach ridge formation.
Soils in dust-influenced arid and semi-arid regions of the American west are thought to form by accretionary processes
where dust influxes increase soil volume and inflationary strain (McFadden, 2013). In this accretion-inflationary mode of
profile development (also known as AIP), dust becomes trapped by vegetation, desert pavement stones, or irregular, rocky
surfaces, then translocates down the soil profile thickening the Bt and Bk horizons. Increases in dust flux during the
Holocene have been linked to a 20 cm rise in ancient Pleistocene desert pavement surfaces on lava flows in the Cima
Volcanic Field in the Mohave Desert, California (McFadden et al., 1987). Thermoluminescence (TL) ages of ~12-13.5 ka
and 5 ka were obtained from Bwk and Av soil horizons, respectively, that are situated on 560 ka lava flows. These ages were
attributed to the continuous supply of eolian materials to the soils during the Late Pleistocene, with an increase in dust flux
during the early to middle Holocene (McFadden et al., 1998).
Lake level reconstructions in our study area suggest that Cave Lake receded from its highstand before ~16,900 years ago
(Duke and King, 2014). Initially these gravels may have been open-work gravels where large pore spaces allowed light
penetration to some depth below the beach surface. As the climate conditions became more arid during and after the
Pleistocene-Holocene transition, dust accumulation would have facilitated the establishment of soil and shrubland vegetation
(t1 in Fig. 16). Root penetration and burrowing animals would have created a zone of bioturbation below the surface. Given
that roots of Great Basin desertscrub species (Spaulding, 1985) can extend several feet below the surface to access water at
depth, this depth of disturbance likely extended up to a meter or more and likely pushed gravels at the surface down to the
depth of sampling. As lakes in the region dried up between ~14,600 and ~9000 years ago, exposed lake bottoms would have
increased the amount of erodible silt and sand available for wind transport, and the Middle Holocene Drought (~8-4 ka)
(Wriston, 2009; Steponaitis et al., 2015) likely maximized dust influx to the beach ridge soils (cf. McFadden et al., 1992).
We infer that accretion-inflationary processes contributed to volume expansion of the soil raising its surface above the
original Late-Pleistocene open-work gravel surface level (t2, Fig. 16). This implies that gravel clasts sampled at ~50 cm
depth in this study, if not bioturbated, would have been several centimeters closer to the original open-work gravel surface
prior to accretion.
The youngest volcanic rock surface ages cluster between ~3.7 ka ~5.4 ka. This suggests that the number of surface gravel
clasts pushed down to the depth of sampling decreased substantially after ~4 ka as the A horizon thickened and the substrate




became less permeable. This implies that by the mid-Holocene the soil surface and the zone of subsurface bioturbation had risen above the level of sampling (t2, Fig. 16).

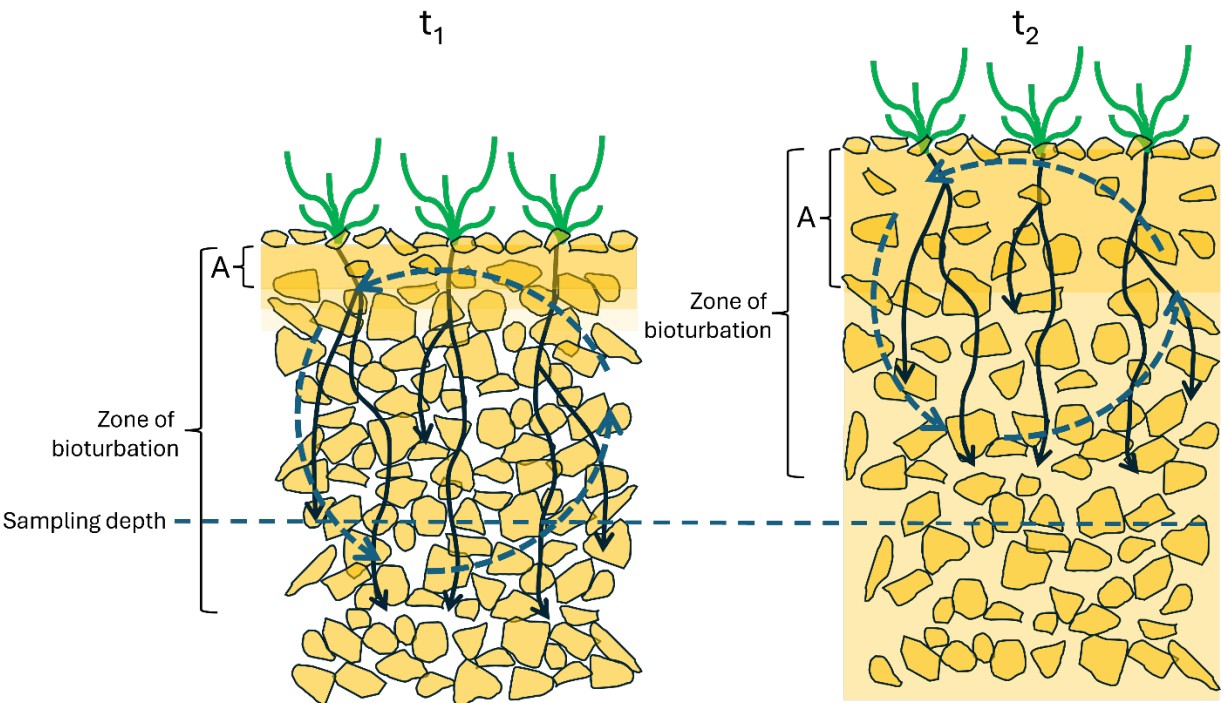

**Figure 16. Schematic illustrating the accretionary-inflationary mode of soil profile development on the beach ridge at site CA21P1. Initial establishment of an A soil horizon after beach ridge abandonment in the late Pleistocene-early Holocene is shown as "t1". Increases in dust influx ~8-4 ka (t2) cause accretion of the A horizon and translocation of fines to depth. As the soil surface rises, the zone of bioturbation rises preventing surface clasts reaching the depth of sampling after ~4 ka.**

### 5.2.4 Significance of volcanic rock plateau ages

Of all age estimates produced from Cave Valley, the pIRIR290 plateau ages from all rocks except Rock 18 most closely approximated the inferred time of the pluvial lake highstand of the region ~16,000 to 18,000 years ago (Fig. 15(b)). In previous studies where cobble-sized rocks are dated, the highest plateau identified near the center of the rock luminescence-depth profile is typically interpreted to record "saturation" (Fig. 1). "Saturation" refers to the area where bleaching has never occurred, and the luminescence signal has reached an equilibrium, where signal loss due to fading (for feldspars) is in equilibrium with signal gained via irradiation. However, in some cases, if the rocks dated are small enough, and if light exposure is long enough, light can penetrate the entire thickness of the rock, leading to a flattening of the plateau. Lehmann et al. (2018) showed that the inflection point of the IR50 luminescence-depth curve migrates to 3-4 mm depth after 20 to 140 years of sun exposure for coarse grained orthogneiss. Ou et al. (2018) found that after 91 days of sun exposure, the PIRIR225 signal was bleached to half of its initial intensity at a depth of ~1.8mm for a dark grey, fine grained indurated sedimentary greywacke. Furhmann et al. (2022) depleted the pIRIR225 signal to a depth of over 2 mm in granite exposed to





the sun for 108 days. Evidence for internal bleaching of IR50 signals in rocks of thicknesses ranging from ~15 to ~23 mm has been shown by Souza et al. (2019) for samples collected from a modern beach and a ~2,000 year old sandy beach ridge in Denmark. These results suggest that for fine-grain, light colored volcanic rocks such as those in our study (Supplementary Section 4), the IR50 signal could be nearly depleted >10 mm into the rock after less than 200 years of exposure. pIRIR signals are known to bleach at a slower rate than IR50 signals, but nonetheless, this begs the question as to whether or not

the Cave Valley beach gravel rocks are small enough such that the pIRIR290 signal was completely depleted throughout their thickness in the swash or near-shore zone of the lake environment prior to burial.

   The MDM age value calculated from all rock plateau ages is 14.68 ± 1.62 (1 σ error) (Fig. 15(b)). At 2 σ, this age overlaps with the time of the C-14 dated highstand of neighboring Lake Valley and suggests that the gravel pIRIR290 plateau ages may serve as more reliable geochronometers for the time of beach formation than the gravel rock surface ages. Such a

scenario could occur if gravels in the swash zone of the lake were sun-exposed for long enough periods to completely deplete their pIRIR290 signals throughout their thickness prior to final emplacement in the beach ridge (cf. Souza et al., 2019). After ridge formation in this scenario, the pIRIR290 signal that accumulated at the center of the gravels during burial may have been less prone to depletion during subsequent brief periods of sun exposure during bioturbation events, which preferentially depleted the signal near the surface of exposed rock surfaces.

**5.2.3 The impact of soil development**

   The Cave Lake highstand at >~16.9 ka (Duke and King, 2014) is thought to have preceded the Lake Coal highstand at 15.9-16.3 cal ka BP (Wriston and Adams, 2020), however despite this, Cave Lake rock surface and internal plateau luminescence ages post-date the highstand of Lake Coal as well as the model-equivalent rock surface ages from the Coal Valley beach ridge (Tables 6 and 8). This is likely due in part to the depth of sampling; at Cave Valley the sampling depth (<0.5 m) was

less than that at Coal Valley (1 m) and clearly within the zone of the A/B soil horizon, thus increasing our chances of sampling bioturbated material. When sampling pluvial lake gravel beach ridges, efforts should be made to sample at depths of ~1 m or greater. Natural exposures of beach ridge sediments such as those sampled at Coal Valley are rare in the Great Basin, and so deeper sampling may require the use of mechanical equipment.

   Dose rates for Cave Valley were calculated assuming that the present geochemistry of the samples have been consistent

during the burial history of the beach ridge. This assumption is problematic given the shallow (<0.5 m) sampling depth within soils that would have developed shortly after beach ridge abandonment and continued to evolve during the Holocene. Accretion-inflationary soil profiles are dynamic, characterized by cumulic growth as well as pedogenic modification. As the soil develops, it increasingly influences infiltration rates, depths of water movement, and thus, rates and processes of carbonate translocation and accumulation for the entire soil (McFadden et al., 1992; 1998). Future luminescence sampling

should ideally focus on primary beach ridge sediments below any soil development until pedogenic effects on time-averaged dose rates are better understood.




## 6 Conclusions

This study examined the feasibility of dating pluvial lake beach ridges using rock surface dating techniques. The geology of the Great Basin is dominated by rock lithologies that pose challenges for luminescence dating and the lack of natural

exposures make sampling below soils and zones of bioturbation difficult. Tests and measurements from two prominent rock types, limestone and volcanic rock, show promise. Select limestone rocks were found to have adequate quantities of detrital sediment preserved near their surfaces for dating. Polymineral extracts from limestones exhibited IR50 signals with low to moderate fading rates and properties suited to SAR. Volcanic rocks exhibited IR50 and pIRIR signals with high fading rates and could only be dated using a high-temperature pIRIR 290 signal with a fading correction. Comparisons between rock age

estimates and independent age control show that both limestone and volcanic rocks are commonly incompletely bleached but also can yield some ages that post-date the time of the pluvial lake highstand. Anomalously young ages from limestones may result from contamination of polymineral extracts with grains that have adhered to the limestone surface in pedogenic carbonate coatings. Young ages obtained from volcanic rocks suggest that most rock surfaces have been exposed to light long after the pluvial lake highstand, likely because of bioturbation. The youngest volcanic rock surface ages provide a

constraint on the timing of pedogenisis and reworking at the sampling depth and perhaps record a climatically-driven phase of soil development. However, the surprising congruency between age-depth profile plateau ages calculated from inside volcanic rocks and independent age control suggests that signals most likely to record the time of beach ridge formation may be preserved in the rock sub-surface.

Rock surface luminescence dating techniques for pluvial lake beach ridges in the Great Basin should be further developed to

build on this work. We recommend preliminary testing of the optical properties of local rock lithologies comprising the site, followed by mechanical excavation to depths below zones of pedogenesis and bioturbation. Experiments should be designed to examine sample collection and preparation methods, gravel bleaching processes in pluvial lake environments and the impact of soil development and bioturbation on sampled sites.

## Data availability

Data generated during the course of this project is available upon request.

## Author contribution

Project conceptualization, fieldwork and sample collection was conducted by CMN and TW. Experiments were designed by CMN with contributions from GTHJ and SH. All experiments and data analyses were conducted by CMN. The manuscript was prepared by CMN with contributions from all authors.



**Competing interests**

The authors declare that they have no conflict of interest.

**Acknowledgements**

Special thanks to Helena Middleton, Saige Howard, Mojtaba Elahifard, Sara Zeitoun, Devon Ardesco and Niko Mastik for help preparing the samples. We thank Dr. Kenneth D. Adams for his geomorphological insights and age controls in Coal

Valley and R. Jake Hickerson, the BLM Basin-and-Range National Monument Archaeologist, who facilitated and encouraged our work.

**Financial support**

This work was supported by funding from the Lincoln County Archaeology Initiative (Round 12) to TW and CMN, NSF Laboratory Technician Support (grant 1914566) and support from Everick Heritage Pty Ltd. to CMN.

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
