# Peer review of "Dating Late Pleistocene pluvial lake shorelines in the Great Basin, USA using rock surface luminescence dating techniques: developing new approaches for challenging lithologies"

_EGUsphere, 2025_

## Referee Comment (RC2)

*Dear authors*

*This is a very large and excellently executed study. The application of SG dating to the rock samples offers a fascinating and innovative perspective. The dose rate and age estimations are carried out with great rigor and attention to detail.*

*The study rightly highlights the important issue of inhomogeneous bleaching history, which is a crucial aspect that could be further emphasized in the abstract. It is clear that one cannot simply assume that dating the entire rock surface yields an accurate age. The SG approach is a valuable and effective method for addressing this challenge.*

*It is a minor limitation that no measurements were taken deeper within the volcanic rocks or from similarly larger samples, which would have provided useful confirmation of whether the luminescence signal is truly saturated or fully bleached before last burial. Additionally, lab-to-field saturation ratios could also offer valuable insight into fading effects.*

*Furthermore, I would have welcomed measurements from both rock types of samples exposed to light at the time of collection, as these would offer further insightful information on the bleaching processes.*

*Overall, this study makes a significant and commendable contribution to the field.*

*Below are some comments to specific lines:*

**L 203: "for only for the limestone"**

*"For" two times*

**L. 206: Traditional?**

*There are several "traditional" preparation methods, such as slicing and grinding, but the choice largely depends on the rock type. You might want to reconsider or clarify the citation here to ensure it accurately reflects the context or specific method being discussed.*

**L. 212: 1. The outer secondary carbonate coatings were filed away with a file or Stylo-style Dremel tool.**

*Could you specify how much material was removed (in mm)?*

**L- 223: During beach ridge formation, light penetrated the outer 2 mm or more of the limestone surface to bleach the signals from detrital quartz and feldspar minerals.**

*The assumption here treats the RSLD sample as if it were sediment, relying solely on surface bleaching. However, since this is a rock surface, there is an opportunity to extract more information by analyzing signal variation with depth. Do you have inner material or a luminescence-depth profile that could support or challenge the assumption of surface bleaching? Otherwise, the unique potential of RSLD compared to sediment may not be fully utilized.*

**L. 244: The polymineral slices were subsequently crushed gently by hand using an agate mortar and pestle and sieved into distinct grain size fractions between 125 and 250 μm for measurement.**

*While it makes sense to extract known grain size fractions, did you assess whether the mechanical crushing process alters the luminescence signal, for example through induced sensitivity changes or signal resetting?*

**L. 278: Aliquots/grains were rejected from further analysis if the recycling ratio was beyond 10% of unity and if recuperation was greater than 5% of the sensitivity-corrected natural signal.**

*Have you assessed whether the rejection of these data points introduces any bias or significantly alters the results?*

**L. 384: Measured-to-given dose ratios were 0.99 ± 0.02, 0.91 ± 0.03 and 0.96 ± 0.03 for Rocks #2, 10 and 18, respectively suggesting that the IR50 SAR protocol is suitable for the Coal Valley limestone samples.**

*However, what about the pIRIR signal? Additionally, have corrections for residuals been applied to the data? Clarification on this would strengthen confidence in the suitability of the pIRIR protocol for these samples.*

**L. 593: As expected, IR50 uncorrected ages are significantly younger than pIRIR290 uncorrected ages (Figs 10 and 11) and this is attributed to the high rate of fading of the IR50 signal as well as the lower bleaching rate of the pIRIR290 signal.**

*However, attributing the age difference to the lower bleaching rate of the pIRIR signal indirectly suggests that the signal may not have been fully reset prior to burial.*

**L. 792: After ridge formation in this scenario, the pIRIR290 signal that accumulated at the center of the gravels during burial may have been less prone to depletion during subsequent brief periods of sun exposure during bioturbation events, which preferentially depleted the signal near the surface of exposed rock surfaces.**

*It would be beneficial to present the age-depth plot with a logarithmic scale on the y-axis. This adjustment may reveal that the IR50 signal is also bleached progressively towards the interior, consistent with the bleaching observed in the pIRIR signal. This bleaching may have been further enhanced after burial. Such evidence would further justify the exclusion of surface slices from the dating analysis.*

**Table 4** *: Could you please include the dose recovery ratios here?*

**Figure 12, 13, 14**: *It is difficult to see the IR50 data. You may consider using a log scale.*

Best Regards

---

## Author Comment (AC2)

[Figure]

Figure A. Dose recovery test results (measured/given dose versus overdispersion) for limestone Rock #2 from Coal Valley (SG data measured with the IR50 signal), and volcanic Rock #18 from Cave Valley (multi-grain aliquot data measured with the PIRIR290 signal). Dose recovery test results have been re-analysed after changing either the recuperation rejection criteria or the recycling ratio rejection criteria. More lenient values are shown in the green, and more conservative or restrictive values are in orange. 'N' refers to the number of single grains/aliquots that were accepted after analysis. The original study (hollow symbol) accepted aliquots or grains with recuperation <5% of the natural signal and recycling ratios <10% of unity.

[Figure]

**Figure 12. Uncorrected (left) and fading-corrected (right) age-depth profiles for volcanic Rocks 4 and 7 from CA21P1. Dashed lines denote slice thickness. Aliquots contain ~125 grains each of the 125-250 μm fraction. Top and bottom of the rock indicated for**

oriented samples. Yellow shading highlights slices used for plateau age calculations. Black dots are pIRIR290 aliquot ages, X symbols are rejected aliquots, and red hollow circles are CDM weighted mean ages for each slice. The y-axes are plotted on a log scale.

[Figure]

Figure 13. Uncorrected (left) and fading-corrected (right) luminescence-depth profiles for volcanic Rocks 11 and 13 from CA21P1. Dashed lines denote slice thickness. Aliquots contain ~125-165 grains each of the 125-250 µm fraction. Yellow shading highlights slices used for plateau age calculations. Black dots are pIRIR290 aliquot ages, X symbols are rejected aliquots, and red hollow circles are CDM weighted mean ages for each slice. The y-axes are plotted on a log scale.

[Figure]

(a) Rock 18 Core 1

(b) Rock 18 Core 1

(c) Rock 18 Core 2

(d) Rock 18 Core 2

(e) Rock 18 Core 3

(f) Rock 18 Core 3

**Figure 14. Uncorrected (left) and fading-corrected (right) luminescence-depth profiles for 3 cores from Rock 18 from CA21P1. Dashed lines denote slice thickness. Aliquots contain ~125 grains each of the 180-250 μm fraction. Yellow shading highlights slices used for plateau age calculations. Black dots are pIRIR290 aliquot ages, X symbols are rejected aliquots, and red hollow circles are CDM weighted mean ages for each slice. The y-axes are plotted on a log scale.**

640

41